# Phagocytosis-initiated tumor hybrid cells acquire a c-Myc-mediated quasi-polarization state for immunoevasion and distant dissemination

Chih-Wei Chou [1,9], Chia-Nung Hung[1,9], Cheryl Hsiang-Ling Chiu [1,9], Xi Tan [1], Meizhen Chen[1], Chien-Chin Chen [2], Moawiz Saeed [1], Che-Wei Hsu[3], Michael A. Liss [4,5], Chiou-Miin Wang [1], Zhao Lai[1], Nathaniel Alvarez [1], Pawel A. Osmulski [1], Maria E. Gaczynska [1], Li-Ling Lin [1], Veronica Ortega[6], Nameer B. Kirma[1], Kexin Xu [1], Zhijie Liu [1], Addanki P. Kumar [1,4,5], Josephine A. Taverna[1,5,7], Gopalrao V. N. Velagaleti [6], Chun-Liang Chen [1,8] ✉, Zhao Zhang [1] ✉ & Tim Hui-Ming Huang [1,5] ✉

While macrophage phagocytosis is an immune defense mechanism against invading cellular organisms, cancer cells expressing the CD47 ligand send forward signals to repel this engulfment. Here we report that the reverse signaling using CD47 as a receptor additionally enhances a pro-survival function of prostate cancer cells under phagocytic attack. Although low CD47-expressing cancer cells still allow phagocytosis, the reverse signaling delays the process, leading to incomplete digestion of the entrapped cells and subsequent tumor hybrid cell (THC) formation. Viable THCs acquire c-Myc from parental cancer cells to upregulate both M1- and M2-like macrophage polarization genes. Consequently, THCs imitating dual macrophage features can confound immunosurveillance, gaining survival advantage in the host. Furthermore, these cells intrinsically express low levels of androgen receptor and its targets, resembling an adenocarcinoma-immune subtype of metastatic castration-resistant prostate cancer. Therefore, phagocytosis-generated THCs may represent a potential target for treating the disease.

Phagocytosis is a process of immune cells engulfing and digesting large foreign pathogens or apoptotic cells for maintaining body homeostasis[1,2]. Macrophages are the main phagocytes that recognize the "find-me" and "eat-me" signals expressed on apoptotic or tumor cells for initiating phagocytosis[3,4]. On the other hand, normal cells or some tumor cells can express "don't-eat-me" signals to repulse phagocytic attacks[5]. The integrin-associated CD47 on the cell surface interacts with SIRPα on macrophages to promote ITIM

---

[1]Department of Molecular Medicine, University of Texas Health Science Center, San Antonio, TX 78229, USA. [2]Department of Pathology, Ditmanson Medical Foundation Chia-Yi Christian Hospital, Chiayi, Taiwan. [3]Department of Pathology, National Cheng Kung University Hospital, College of Medicine, National Cheng Kung University, Tainan, Taiwan. [4]Department of Urology, University of Texas Health Science Center, San Antonio, TX 78229, USA. [5]Mays Cancer Center, University of Texas Health Science Center, San Antonio, TX 78229, USA. [6]Department of Pathology and Laboratory Medicine, University of Texas Health Science Center, San Antonio, TX 78229, USA. [7]Department of Medicine, University of Texas Health Science Center, San Antonio, TX 78229, USA. [8]Biobehavior Laboratory, School of Nursing, University of Texas Health Science Center, San Antonio, TX 78229, USA. [9]These authors contributed equally: Chih-Wei Chou, Chia-Nung Hung, Cheryl Hsiang-Ling Chiu. ✉e-mail: chenc4@uthscsa.edu; zhangz3@uthscsa.edu; huangt3@uthscsa.edu

phosphorylation, which activates SHP phosphatases to cause cytoskeletal remodeling and membranous motility needed for phagocytosis[5,6]. In addition to CD47-SIRPα, a growing number of "don't-eat-me" signals have recently been identified with anti-phagocytosis functions, including CD24-Siglec 10, PD-L1-PD-1, MHC-1-LILRB1, and APMAP-GPR84[7–10]. Since tumors are usually replete with macrophages, targeting these macrophage checkpoints with inhibitors represents an attractive strategy for cancer immunotherapy[5,11].

While the forward action of CD47-SIRPα to repress phagocytosis is well studied, there is presently limited information regarding the effect of the reverse signaling[12]. There is a likelihood that the reverse signaling is activated on the entrapped cells[13,14]. Upon macrophage contact, the ligand CD47 could function as a receptor of SIRPα to trigger downstream signaling cascades in a cell. Without macrophage contact, CD47 signaling can also be activated in response to soluble SIRPα known to regulate multiple cellular functions, including proliferation and migration[13,15]. Whereas targeting the CD47-SIRPα interaction restores phagocytic activities, the strategy is sometimes suboptimal due to the possibility that tumor cells exploit the reverse signaling mechanism for survival. Therefore, additional studies to evaluate this reverse signaling are critical to increasing our understanding and improving the designs of macrophage checkpoint inhibitors.

Motivated by the need to investigate the function of CD47-SIRPα reverse signaling, we conduct phagocytosis assays in prostate cancer cells expressing different levels of CD47. Low CD47-expressing cancer cells are more prone to phagocytosis; however, the phagocytic clearance is sometimes thwarted due to reverse signaling activities that support anti-apoptotic effects on these cells. The procrastination of complete digestion allows macrophages to obtain and assimilate genetic material from the entrapped cancer cells, resulting in the formation of tumor hybrid cells (THCs). These cells exhibit characteristics of parental macrophages and cancer cells. Furthermore, we investigate how THCs acquire c-Myc from parental cancer cells to upregulate M1- and M2-like macrophage polarization genes. This quasi-polarization phenotype allows THCs to mimic diverse macrophages to evade immunosurveillance in the tumor microenvironment. We also explore whether these cells gain castration-resistant attributes through transcriptomic reprogramming and acquire epithelial-mesenchymal plasticity in the bloodstream for distant dissemination. In this study, we provide experimental and clinical evidence to demonstrate a plausible cause and consequence of prostate THCs.

## Results

### Low CD47-SIRPα signals render incomplete phagocytosis and THC formation

*CD47* overexpression is thought to be associated with advanced cancers[16–18]. Contrastingly, we observed that metastatic prostate tumors frequently expressed lower *CD47* levels, compared with primary tumors and benign hyperplasia samples (Fig. 1a)[19–21]. This low expression was also linked to shorter biochemical recurrence (BCR)-free survival in the TCGA cohort (Fig. 1b). To determine if low *CD47* activities influence the interaction between macrophages and cancer cells, we conducted phagocytosis assays with prostate cancer cell lines expressing different CD47 levels (Fig. 1c). EGFP-labeled cancer cells were permeated with the Tag-it-Violet (TiV) dye and then co-cultured with primary monocytes, U937 monocytes, or their respective differentiated macrophages. Flow cytometry analysis showed that cancer cells being engulfed and digested by macrophages displayed CD86⁺/TiV⁺/EGFP⁻. However, a CD86⁺/TiV⁺ subpopulation was found to express diverse levels of EGFP⁺, likely attributed to various extents of incomplete phagocytosis (Fig. 1d; see also time-lapse imaging in Supplementary Fig. 1a). Immunofluorescence imaging confirmed that this incomplete digestion of a cancer cell inside a monocyte/macrophage led to THC formation (Supplementary Fig. 1b, c).

In subsequent analysis, we gated putative THCs as those co-expressing macrophage CD86 and cancer cell-labeled EGFP markers (Supplementary Fig. 1d). An increase in THC formation was observed in cancer cells co-cultured with differentiated macrophages relative to those with monocytes, likely reflecting a higher ingestion capacity of the former than the latter and thus increasing the chance of forming THCs (Fig. 1e, Supplementary Fig. 1e). Moreover, THC formation was increased when macrophages derived from the oncogenic U937 monocyte cell line were used in co-culture relative to those differentiated from primary monocytes (Fig. 1e). We speculated that phagocytosis could be less robust in U937 macrophages than in primary macrophages, resulting in increased incomplete digestion and THC formation. Compared to U937 macrophages, primary macrophages appeared to digest C4-2 cancer cells more efficiently and led to a more rapid decrease in the proportion of these cells, reducing from 40% at 2 h to 2% at 72 h in co-culture (Supplementary Fig. 1f). Our additional co-culture assay revealed that the main source of THCs was likely derived from stem-like side populations of a cancer cell line, supporting the previous findings (Supplementary Fig. 1g)[22,23].

When comparing the cancer cell lines, we found that the overall efficiency of forming THCs was more pronounced in low CD47-expressing C4-2 cells relative to high CD47-expressing 22Rv1 or DU145 cells co-cultured with macrophages (Fig. 1e). Co-culture experiments using *CD47* knockin or knockdown cells partially decreased or increased THC formation, respectively (Fig. 1f, g). Furthermore, THC formation in co-culture was increased by pre-blocking CD47-SIRPα signals in cancer cells with an antibody (Supplementary Fig. 1h, i). The interference of two other "don't-eat-me" signals, CD24 and PD-L1, did not exert apparent effects, further highlighting the critical role of weak CD47-SIRPα signals in promoting THC formation (Supplementary Fig. 1j, k).

Whereas the forward action of CD47-SIRPα is to repel phagocytosis initiated by macrophages, we speculated that the reverse signaling supports pro-survival activities of the entrapped cancer cells. Indeed, we found that THCs displayed higher expression levels of pro-survival BCL-2 relative to parental macrophages or cancer cells (Fig. 1h–j). Increased BCL-2 expression might enhance anti-apoptotic activities, thus preventing the complete digestion of the entrapped cancer cells inside macrophages. To mimic this reverse interaction, soluble SIRPα was then added to cell cultures. Despite BCL-2 levels being enhanced in parental cancer cells and macrophages, the induction was again more pronounced in putative THCs (Supplementary Fig. 1l, m). The binding of free SIRPα to CD47, confirmed by a proximity-ligation assay, partially de-repressed the expression of BCL-2 and Lamin A/C in C4-2 cells undergoing TNFα-mediated apoptosis (Fig. 1k, l). Conversely, the expression of apoptotic Annexin V was decreased in these cells (Fig. 1m). As a result, cell viability was modestly enhanced, showing a decreased number of dead cells from 51 to 46% (Fig. 1n, Supplementary Fig. 1n). Taken together, we propose a model whereby strong CD47-SIRPα signaling not only represses macrophage encroachments but also enhances the survival of a cancer cell under this phagocytic attack (Fig. 1o). Whereas low CD47-expressing cancer cells allow phagocytosis, their reverse signaling interferes with this process by partially activating a pro-survival function, leading to incomplete digestion of the entrapped cells and THC formation (Fig. 1p).

### THCs evade immunosurveillance to gain a survival advantage in the host

To assess the tumorigenicity of THCs, we subcutaneously inoculated co-culturally enriched THCs or parental control cells to Nu/Nu mice. Although defective in the adaptive immune system such as T cells[24], these mice retain macrophage lineages and other myeloid cell populations suitable for investigating the extent of their infiltration into xenograft tumors. Three of the 10 mice transplanted with C4-2 THCs showed exponential growth within 10 weeks (Supplementary Fig. 2a).

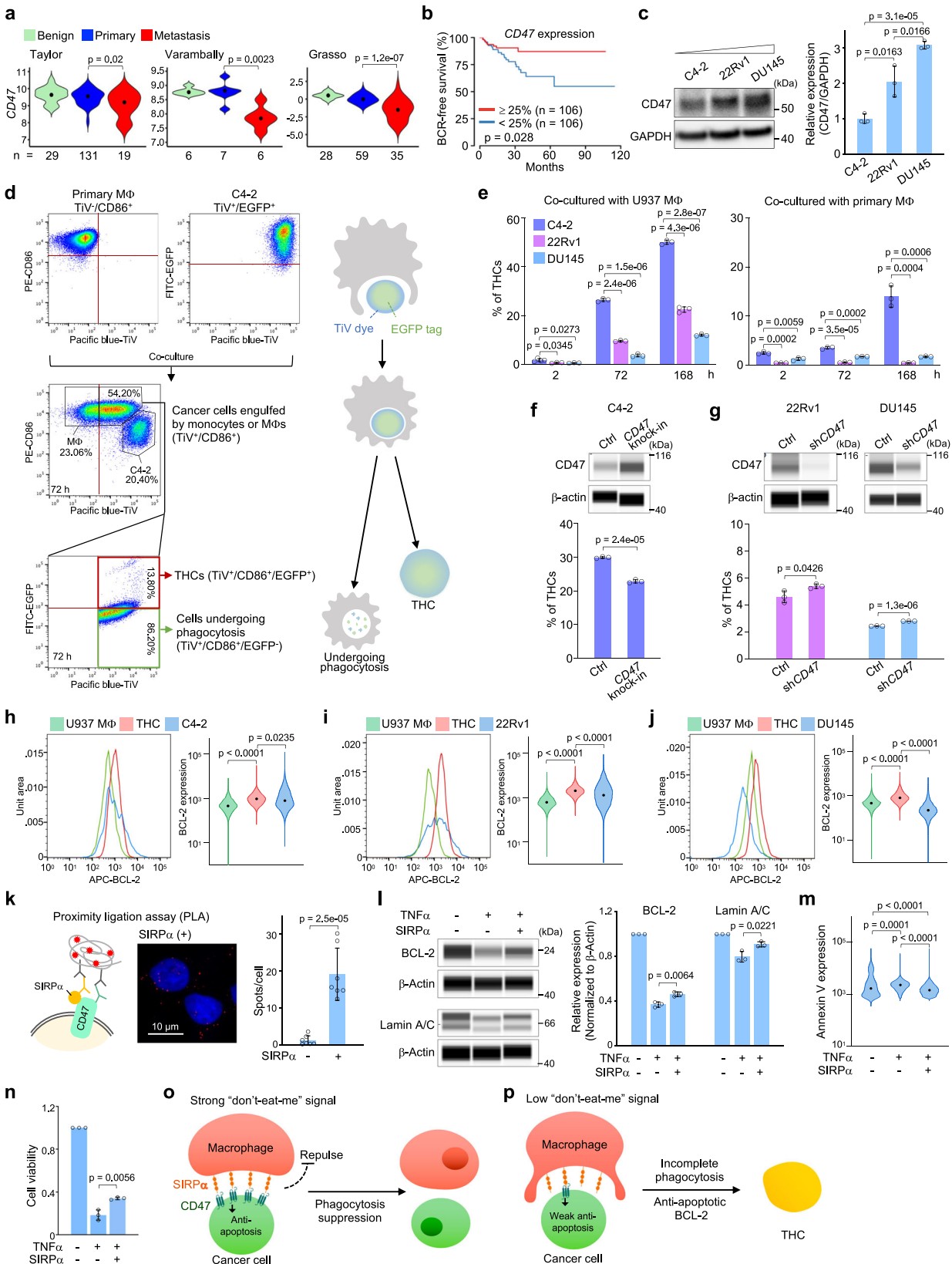

Mice inoculated with parental cancer cells all developed tumors, albeit at slow rates with flattened curves (Supplementary Fig. 2b). Inoculated parental macrophages had negligible growth in mice, except for one with a visible lump, possibly a lymphoma developed from residual U937 monocytes non-responsive to the stimulation of phorbol-myristil-acetate (PMA) (Supplementary Fig. 2c)[25].

To determine the influence of THCs on the host microenvironment, we performed spatial gene profiling on two xenograft tumors derived from parental cancer cells and co-culturally enriched THCs, respectively (Fig. 2a). A tumor section was placed onto a slide area containing ~5000 barcoded spots, each capturing 1-10 cells[26]. The section was permeabilized to release mRNAs for cDNA sequencing and

**Fig. 1 | Low CD47 facilitates THC formation between macrophages and cancer cells. a** CD47 expression levels in benign prostatic hyperplasia, primary, and metastatic tumors analyzed from three published prostate cancer datasets[19–21]. **b** Biochemical recurrence (BCR)-free survival of the prostate cancer patients in the TCGA cohort. **c** Western blotting analysis of CD47 expression levels in three human prostate cancer cell lines. **d** A representative example of flow cytometry analysis of THCs in human primary macrophage and C4-2 cell co-culture model. Some CD86+ macrophages acquire Tag-it-Violet (TiV) dyes through engulfing cancer cells in co-culture. The cells maintained EGFP expression in this TiV+/CD86+ population were further identified as THCs. **e** Percentage of THCs in different co-culture models at different time points. **f, g** Percentage of THCs in 3-day co-cultures of U937 macrophages and *CD47* overexpressed (**e**) or knockdown (**g**) cells. **h–j** Flow cytometry histograms and the corresponding violin plots showing BCL-2 expression in U937 macrophages and C4-2 (**h**), 22Rv1 (**i**), or DU145 (**j**) cells co-culture model. **k** Left: schematic diagram of proximity-ligation assay (PLA). Right: PLA signals of C4-2 cells

with or without SIRPα stimulation. The dots represent mean values of total PLA counts per cell (*n* = 7 images/group, 180 cells analyzed). **l–n** Effects of C4-2 cells unstimulated or stimulated by TNFα or TNFα combined with SIRPα. Capillary Western immunoassay (WES) showing BCL-2 and Lamin A/C expression (**l**), violin plots showing Annexin V expression (**m**), and bar graph showing cell viability (**n** each dot represents the mean for each independent experiment). **o** Schematic diagram showing strong "don't-eat-me" signals effectively inhibit phagocytosis. **p** Schematic diagram showing low "don't-eat-me" signals allow macrophages to engulf cancer cells but partially increase the pro-survival function of cancer cells to escape from complete phagocytosis, subsequently resulting in THC formation. Data are the mean ± SD. *P*-values were determined using a two-sided unpaired *t*-test (**a, c, e–l, n**), a log-rank test (**b**), and a one-way ANOVA followed by Tukey test (**m**). Three independent experiments were carried out in (**c, e–g, l, n**). Source data for **c, e–n** are provided as a Source Data file.

data analysis. t-distributed stochastic neighbor embedding (t-SNE) was run on the normalized filtered feature-barcode matrix to reduce the number of feature (gene) dimensions. Graph-based clustering then identified ten major geographical regions (8–38%; labeled #1–10) and three minor regions 1a (0.3%), 3a (3.8%), and 8a (1.2%) located on the parental and hybrid tumor sections (Fig. 2b, c, Supplementary Fig. 2d, e). We excluded genes highly conserved between human and murine cells and identified 429 human and 319 murine differentially expressed genes among these regions (Supplementary Fig. 2f). Regions 1a and 8a in the hybrid tumor shared identical gene signatures as regions 1 and 8 in the parental tumor, respectively. These minor regions likely originated from residual parental cancer cells not engulfed by macrophages in the co-culture. On the other hand, region 3a in the parental tumor had the same gene signatures as the main region 3 of the hybrid tumor. This minor region was later shown to have enriched murine gene signatures, suggesting that it contained naturally formed THCs through fusions between human cancer cells and murine macrophages.

We observed that the hybrid tumor usually expressed higher human genes such as *VIM* and lower murine genes such as *Vim* compared to the parental tumor (Fig. 2d, e, Supplementary Fig. 3a–d). When dividing expression profiles into individual geographic regions, we observed that parental tumor regions 1, 5, 6, and 8 contained lower levels of human genes and higher levels of murine genes than those regions 3, 4, and 7 of the hybrid tumor (Supplementary Fig. 3e, f).

The AUCell analysis of 14 oncogenic MSigDB genesets revealed increased expression of human MYC targets V1 geneset in hybrid tumor regions relative to parental tumor regions, while the expression of the remaining human and murine genesets was not overtly different among these regions (Supplementary Fig. 3g, h)[27]. However, increased expression of eight human immune genesets was seen in the hybrid tumor relative to the parental tumor (Fig. 2f). Among these pathways, increased expression of human M1-like (*n* = 25) and M2-like (*n* = 21) genes for macrophage polarization were the most apparent in hybrid tumor regions (Fig. 2g, h, Supplementary Table 1). Concordantly high expression of these genes was also observed in >75% of the regions, suggesting the dual presence of human macrophage lineages in the hybrid tumor (Fig. 2i). Contrarily, the parental tumor mirrored the opposite scenario, showing enriched murine M1-like (*n* = 25) and M2-like (*n* = 18) macrophage genes (e.g., region #1, 5, 6, and 8 in Fig. 2j, k, Supplementary Table 1). While concordantly high expression of the murine counterparts was observed in ~25% of the parental tumor, their patterns pointed to diverse macrophage populations infiltrating into different regions (Fig. 2l). These spatial mapping data suggest that THCs can benefit from the quasi-macrophage characteristics as camouflage to evade the surveillance of murine macrophages or other myeloid cells, including pro-tumorigenic myeloid-derived suppressor cells (MDSCs) and anti-tumorigenic natural killer (NK) cells (Supplementary Fig. 3i, j)[28,29].

## THCs acquire c-Myc as an unexpected immunoevasive driver for M1/M2 gene transcription

Since the expression of MYC targets V1 geneset was increased in the hybrid tumor (Supplementary Fig. 3h), we aimed to determine if this upregulation is attributed to the enhanced activity of the c-Myc transcription factor. Indeed, we found increased c-Myc expression in THCs relative to parental cancer cells or macrophages in co-culture (Fig. 3a, b). This increase was shown to be partly due to copy-number gains of the *MYC* locus originating from parental cancer cells but not from parental macrophages by immunoFISH assays (Fig. 3c). Our spatial mapping data further suggest that increased c-Myc was correlated with elevated expression of M1- and M2-like genes in the hybrid tumor relative to the parental tumor (Supplementary Fig. 4a). To substantiate the finding, we conducted c-Myc ChIP-qPCR of nine representative loci in two hybrid tumors, parental macrophages, and parental cancer cells (Supplementary Table 2). Increased c-Myc binding to their promoters was observed in THCs relative to that in parental macrophages or cancer cells (Fig. 3d-*upper*). The binding was associated with the increased binding of RNA polymerase II for gene transcription, compared to that of IgG controls (Fig. 3d-middle-lower). Nevertheless, the binding patterns of RNA polymerase II to *HLA-E*, *PPARG*, and *CD86* loci in THCs were not overtly different from those in parental macrophages. Their transcription could be preferentially driven by IFN-γ/pSTAT1 or JAK/pSTAT6 in macrophages[30–32]. However, c-Myc was later recruited to these loci to upregulate their expression in THCs (Fig. 3e).

To confirm the regulatory role of c-Myc, we derived five cell lines from hybrid tumors, termed CWC1–5, and authenticated their dual macrophage-epithelial characteristics by immunofluorescent and karyotypic analyses (Supplementary Fig. 4b–d). These THC lines exhibited proliferative capacity, forming cell spheres in vitro similar to those previously reported (Supplementary Fig. 4e)[33,34]. Then, we generated CWC cells carrying *MYC* shRNA knockdown (KD) (Fig. 3f). This shRNA KD resulted in the downregulation of M1- and M2-like genes, thus suggesting the immunoregulatory role of c-Myc in THCs (Fig. 3g). To determine if this immunoregulatory function affects effector T cells, we co-cultured CWC cells carrying control or *MYC* shRNA KD with human peripheral blood mononuclear cells (PBMCs) subsequently stimulated with mitogenic lectins to activate T cells. Flow cytometry analysis was performed to identify cytotoxic CD8+ T cells, regulatory CD25+/CD4+ T cells, and naive CD25−/CD4+ T cells after the co-culture (Supplementary Fig. 4f). The proportion of cytotoxic T cells showed a twofold increase in PBMCs co-cultured with control CWC cells relative to PBMCs only (Fig. 3h-left). This T cell population was further increased when co-culturing with CWC cells carrying *MYC* shRNA KD. An increase (60%) of regulatory T cells occurred in PBMCs co-cultured with control CWC cells relative to PMBCs only (Fig. 3h-middle). Nevertheless, this induction was reduced by exposing PBMCs to cells carrying *MYC* shRNA KD. The shRNA KD had no effect on naive

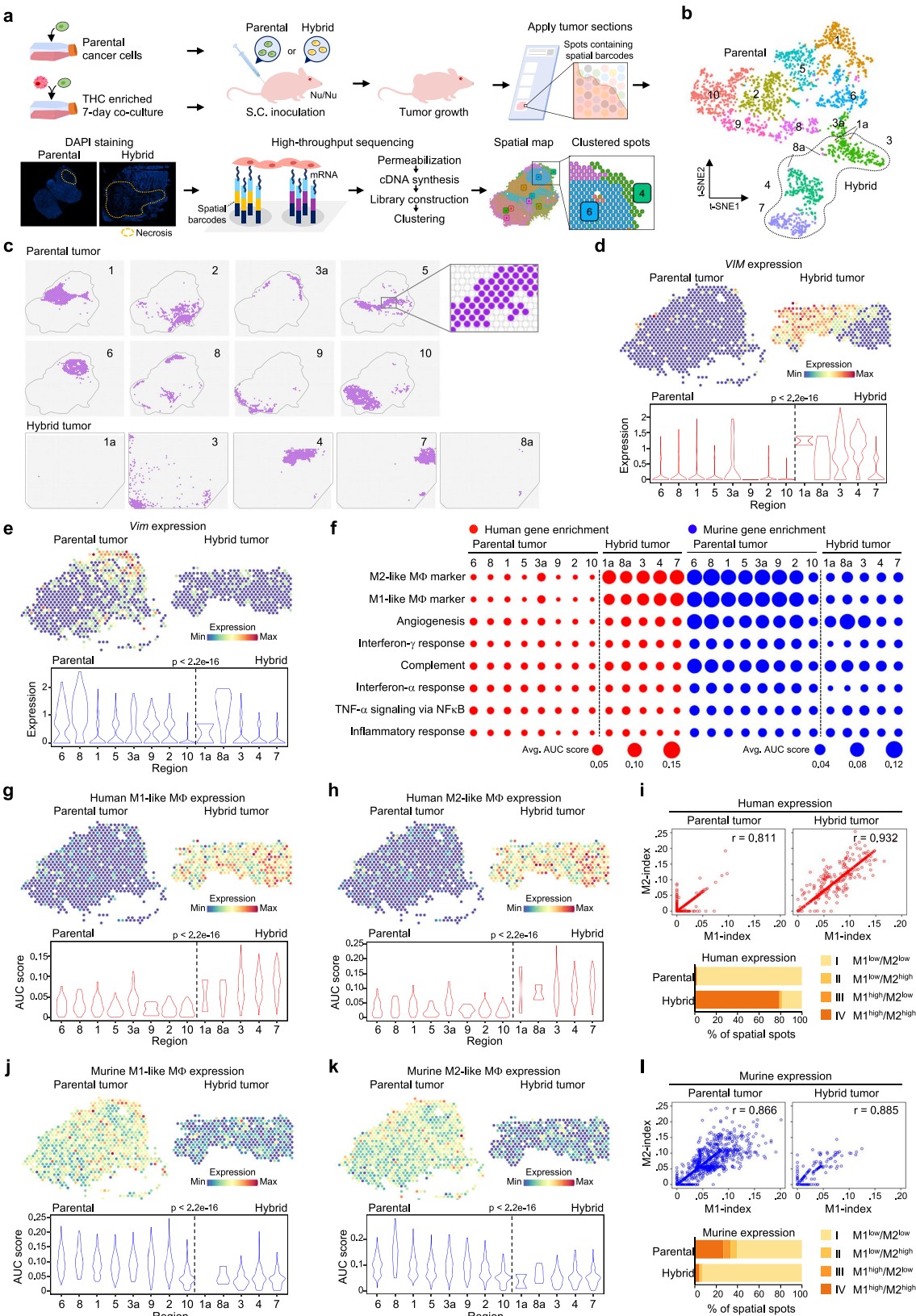

CD25[-]/CD4[+] T cells (Fig. 3h-right). Collectively, this experiment suggests that *MYC* knockdown may unmask immunoevasive characteristics of CWC cells, fostering the involvement of cytotoxic T cells to initiate an anti-tumorigenic microenvironment[35]. In addition, macrophage infiltration was repressed by pre-treating THCs with a c-Myc inhibitor in transwell migration assay (Supplementary Fig. 4g). Based

on the finding, we propose a model in which the transcription of macrophage polarization genes is normally regulated via pSTAT1 or pSTAT6 in parental macrophages[30,36], while the same loci are not expressed in parental cancer cells (Fig. 3i-left). However, THCs exploit overexpressed c-Myc derived from cancer cells to upregulate M1- and M2-like loci originating from macrophages (Fig. 3i-right). Overall,

**Fig. 2 | Spatial transcriptomics analysis unveils an increased expression of human immune markers in hybrid tumor. a** Workflow of spatial transcriptomics analysis of xenograft C4-2 and enriched THCs tumors. See detailed experimental protocol in the Methods. **b** t-SNE plot depicting ten main and three minor regions in parental and hybrid tumors. Note: regions 1, 3, and 8 were shared in both parental and hybrid tumors. **c** The corresponding spatial spots of each region in the parental and hybrid tumor sections. **d, e** The expression of human (**d**) or murine (**e**) vimentin gene. Upper: representative spatial heatmap in the parental tumor regions 1, 5, and 6, and hybrid tumor regions 4 and 7. Lower: violin plot showing the expression in each region. **f** Relative enrichment of eight immune-related genesets across different regions by AUCell analysis. The dot size represents the average gene signature score over all spots in each region. **g, h** The expression of human M1-like (**g**) and M2-like (**h**) macrophage genesets. Upper: representative spatial heatmap in the parental tumor regions 1, 5, and 6, and hybrid tumor regions 4 and 7. Lower: violin plot showing the expression in each region. **i** Upper: correlation of human M1-like and M2-like index values, each scatter dot represents a spatial spot. Lower: proportion of four categories stratified based on the upper quartile of the human M1- and M2-like index values. **j, k** The expression of murine M1-like (**g**) and M2-like (**h**) macrophage genesets. Upper: representative spatial heatmap in the parental tumor regions 1, 5, and 6, and hybrid tumor regions 4 and 7. Lower: violin plot showing the expression in each region. **l** Upper: the correlation of murine M1- and M2-like index values. Lower: proportion of four categories stratified based on the upper quartile of murine M1- and M2-like index values. *P*-values were determined using a two-sided Spearman's rank correlation (**d, e, g, h, j, k**).

c-Myc can be considered an immunoevasive driver in THCs, not just an oncogenic driver per se[37].

## THCs exert migration potential for organ metastasis

To determine the impact of THCs on metastasis, we further isolated cells from the spine or femur fragment of xenograft hosts for bone-in-culture array (BICA; Fig. 4a)[38]. Immunofluorescence analysis identified ~1% of putative THCs in BICA, suggesting these cells migrated from subcutaneous injection sites to the bone (Fig. 4b, c). Nevertheless, macrometastatic lesions to other organs were rarely observed in the initial transplantations of THCs. Additional re-transplantations showed the aggressive growth of hybrid tumors not only in injection sites but also in distant organs (Fig. 4d, e). Then, we conducted single-cell proteomic cytometry by time-of-flight (CyTOF) to characterize cellular components of a metastatic lesion (Fig. 4a, Supplementary Table 3). Murine cells were gated to identify CD90.2$^+$ fibroblasts, CD31$^+$ endothelial cells, and CD45$^+$ immune cells, while human THCs were selected based on their negative staining with these murine markers (Fig. 4f). We found that the majority (94.6%) of 19,257 cells analyzed in a metastatic lesion were of human origin (Fig. 4g). There were very low recruitments of murine fibroblasts (4.7%), endothelial cells (0.6%), or immune cells (0.1%) to the lesion site, similar to the prior finding of primary hybrid tumors with very little infiltration of host cells. PhenoGraph clustering overlaid on a uniform manifold approximation and project (UMAP) plot revealed 15 subpopulations of metastatic THCs (Fig. 4h). When aligning these subpopulations by c-Myc expression levels, we observed that elevated c-Myc was associated with the upregulation of epithelial, mesenchymal, M1-like, and M2-like macrophage markers (Fig. 4i). In line with our previous finding (Supplementary Fig. 4a), elevated c-Myc was also concordant with a high expression of M1- and M2-like markers in THCs (Fig. 4j). When categorizing these THCs into four groups based on mean M1 and M2 indices, we found that elevated expression of both epithelial and mesenchymal markers was associated with concordantly high expression of M1- and M2-like markers (i.e., Category IV: M1$^{high}$/M2$^{high}$ in Fig. 4k-m). The finding suggests that the c-Myc-mediated quasi-polarization phenotype of THCs can boost their epithelial-mesenchymal plasticity and adapt to the host immune microenvironment.

## THCs arise de novo in primary and metastatic lesions

The above findings were based on co-culture-generated THCs. Hence, we asked whether THCs also occur naturally in the tumor microenvironment. Immunohistochemical (IHC) analysis revealed that THCs, positive for macrophage F4/80 and epithelial pan-CK, were present in 1–2% of subcutaneous tumor after inoculating low CD47-expressing C4-2 cells into immunocompromised Nu/Nu mice (Fig. 5a). This low incidence of THC formation was probably a result of less infiltration of host macrophages into subcutaneous tumor xenografts. However, THCs became highly prevalent in a metastatic liver lesion when C4-2 cells invaded the tissue through the bloodstream in tail vein-injected Nu/Nu mice model (Fig. 5b-left). Compared to normal

tissue, an adjacent uninvolved liver site displayed intense infiltration of singular F4/80$^+$ macrophages with a probable F4/80$^+$pan-CK$^+$ THC cluster (Fig. 5b-middle-right). Interestingly, THC clusters became apparent in more than 30% of a metastatic liver site (Fig. 5c). We suggest that this increased macrophage infiltration additionally contributes to THC formation in low CD47-expressing cancer cells.

Nevertheless, the above finding still does not reflect a true de novo event in a normal setting with natural immune responses. Therefore, two syngeneic prostate cancer cell lines, RM-1 and TRAMP-C2, were used for additional tail vein injections into immunocompetent mice. RM-1 cells had high Cd47 expression and showed a lower rate of THC formation in vitro compared to low Cd47-expressing TRAMP-C2 cells (Supplementary Fig. 5a–h). Then, we compared their rates of THC formation in vivo. Although invading RM-1 cells formed large lesions in the lung, the incidence of THC formation was low (~2%), reflecting the in vitro finding (Fig. 5d–f). In contrast, invading TRMAP-C2 cells developed micrometastasis in the lung but displayed THCs in ~30% of the lesions (Fig. 5g–i).

To further investigate THCs in human prostate cancer, we conducted in silico analysis of single-cell RNA-seq datasets processed from a prostate cancer cohort (Supplementary Fig. 6a)[39]. Around 0.7% of 22,084 cells were identified as putative THCs based on co-expression of epithelial and macrophage genes (Supplementary Fig. 6b, c). AUCell analysis again revealed that the expression of M1- and M2-like genesets were increased most apparently in THCs relative to epithelial cells (Supplementary Fig. 6d). Upregulation of *MYC* was also observed in these naturally occurring THCs, supporting the observation of our xenograft models (Supplementary Fig. 6e). Furthermore, increased THC formation tended to occur in tumors with comparatively lower *CD47* levels than those with higher levels (Supplementary Fig. 6f–h).

IHC analysis was further conducted to visualize THCs in 94 prostate tumors based on their double staining features that differed from single stained pan-CK$^+$ tumor cells (Fig. 5j-left). These cells were also distinctive from CD68$^+$ macrophages or those ring-like phagocytes containing engulfed cancer cells (Fig. 5j-right). When classifying tissue sections into Category I-III based on the number of THCs, we found remarkable inter- and intra-tumor variations in tumors (Fig. 5k, l, Supplementary Fig. 6i, Supplementary Tables 4 and 5). Category III tissues showing more THCs were found in metastatic tumors relative to primary tumors, those with higher Gleason scores, and patients of older ages, but not high PSA levels (Fig. 5m–p).

## THCs resemble a subtype of metastatic castration-resistant prostate cancer (mCRPC)

The implication that THCs are linked to metastatic prostate cancer led us to examine whether AR signaling is deregulated in these cells. c-Myc overexpression, known to antagonize AR in prostate cancer[40], might diminish its downstream signaling activities in THCs. Indeed, our spatial profiling revealed that the expression levels of *AR* and its cooperating *FOXA1*[41] were not detectable in the hybrid tumor, although their expression levels were found in 20–40% of the parental tumor (Fig. 6a, b). Very low AR signaling activities led to little or no AR-

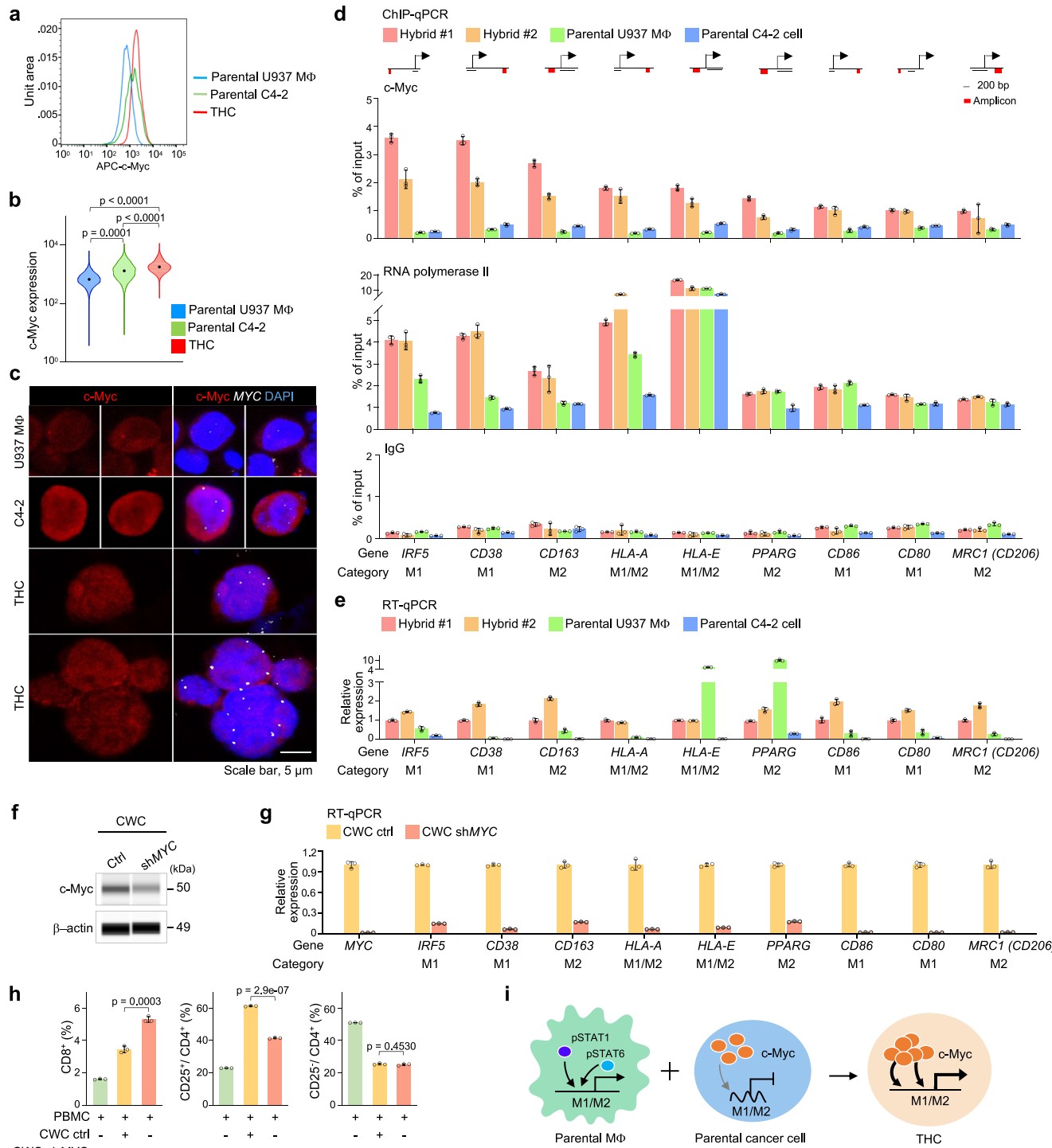

**Fig. 3 | Elevated c-Myc obtained from parental cancer cells upregulates M1- and M2-like macrophage genes in THCs. a** Flow cytometry histogram of c-Myc expression in co-cultured macrophages, cancer cells, and THCs. **b** The corresponding violin plot of a. **c** Representative immunoFISH images of *MYC* loci (white dots) and c-Myc expression (red) in parental C4-2 (*n* = 58 cells), U937 macrophages (*n* = 54 cells), and THCs (*n* = 62 cells). Scale bar, 5 μm. **d** ChIP-qPCR of c-Myc (*upper*), RNA polymerase II (*middle*), or IgG (*lower*) binding to the promoter regions of M1- and M2-like genes of two independent hybrid tumors, parental macrophages, and parental C4-2 cancer cells (*n* = 3 technical repeats). The schematic diagram of the promoter region of each gene was shown. Red bars, primer amplicon sites. Scale bar, 200 bp. **e** Real-time RT-qPCR of the expression of M1- and M2-like genes in two independent hybrid tumors, parental macrophages, and parental C4-2 cancer cells (*n* = 3 technical repeats). **f** WES showing c-Myc

expression in the THC line created from hybrid tumor, CWC cells, infected with scramble or sh*MYC* lentivirus. **g** Real-time RT-qPCR of the expressions of M1- and M2-like genes in CWC cells infected with scramble or sh*MYC* lentivirus (*n* = 3 technical repeats). **h** The population of CD8⁺ T cells, regulatory CD25⁺/CD4⁺ T cells, or naive CD25⁻/CD4⁺ T cells in human peripheral blood mononuclear cells (PBMCs) after co-cultured with CWC cells infected with scramble or sh*MYC* lentivirus (*n* = 3 independent experiments). **i** Proposed model depicting the transcription of c-Myc-mediated M1- and M2-like genes in THCs. Data are the mean ± SD. *P*-values were determined using a one-way ANOVA followed by Tukey test (**b**) and a two-sided unpaired *t*-test (**h**). Data represent two independent experiments (**d, e, g**). Source data for **b, d, e, g, h** are provided as a Source Data file.

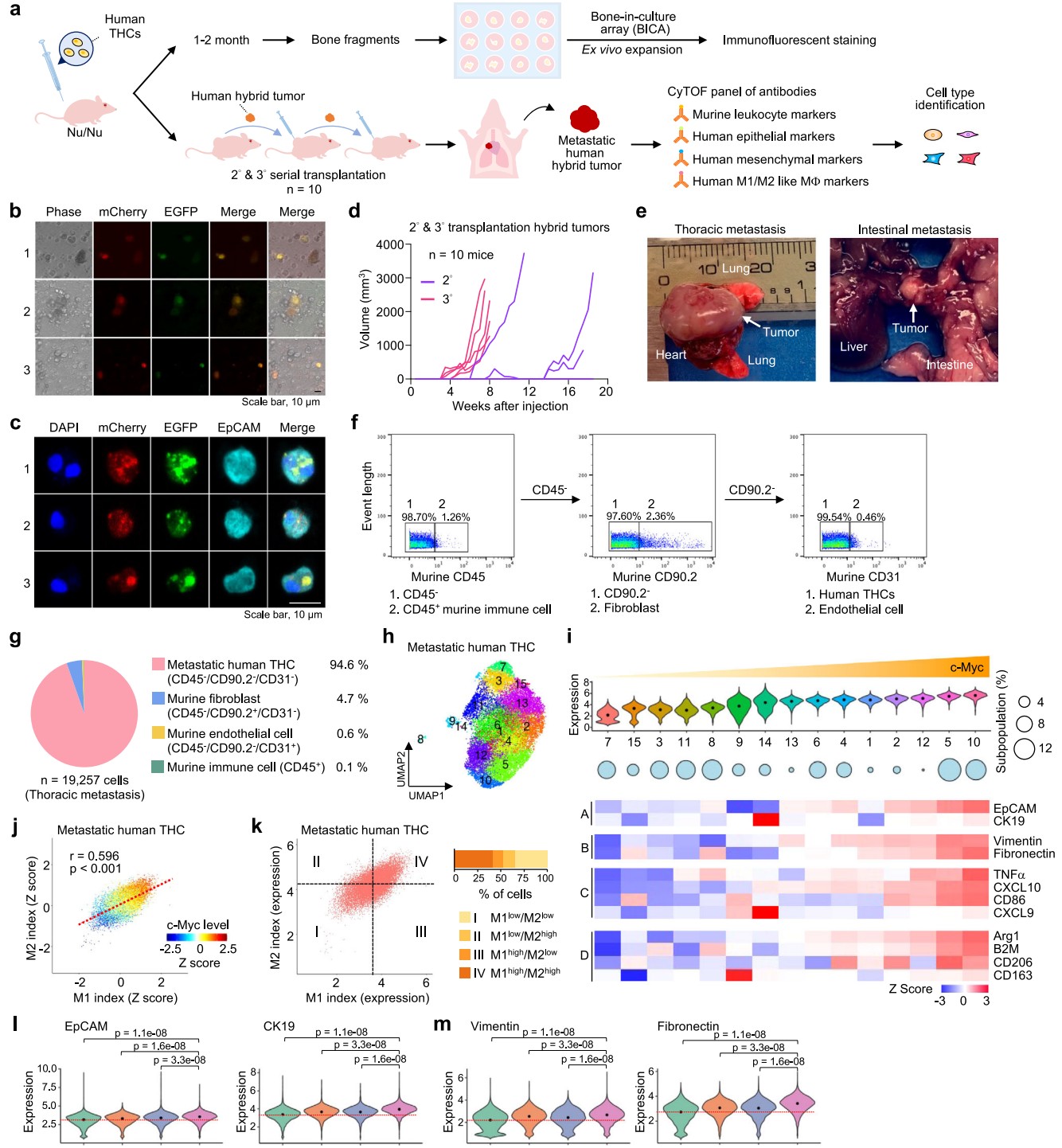

**Fig. 4 | Increased c-Myc is associated with upregulated M1- and M2-like macrophage genes and epithelial-mesenchymal plasticity in metastatic human hybrid tumor. a** Schematic diagram of serial transplantation of human hybrid tumor in mice. THCs in the femur and spine were enriched through bone-in-culture array (BICA). A metastatic tumor found after serial transplantation was analyzed by single-cell proteomic cytometry by time-of-flight (CyTOF). **b**, **c** Representative inverted microscope images (b, *n* = 5 mice) and immunofluorescence images (c, *n* = 2 mice) of BICA cells. A bi-nucleated cell was shown in cell #1 in panel c. Scale bar, 10 μm. **d** Growth curves of the secondary (purple, *n* = 5 mice) and tertiary (red, *n* = 5 mice) transplanted hybrid tumors. **e** A visible metastatic lesion in the thoracic cavity of one mouse transplanted with hybrid tumors and an intestinal lesion in another xenotransplanted mouse. **f**–**m** The thoracic metastatic tumor was analyzed by CyTOF. **f** CyTOF gating strategy to identify metastatic human THCs and host

cells in the thoracic metastatic tumor. **g** Pie chart showing the proportions of cell types in the thoracic metastatic tumor. **h** UMAP plot depicting 15 subpopulations of metastatic human THCs. **i** *Upper*: the 15 subpopulations identified in (**h**) were aligned by increased expression levels of c-Myc. Middle: circle plot depicting the subpopulation sizes. Lower: heatmaps showing the expression of epithelial (A), mesenchymal (B), and M1- (C) and M2-like (D) macrophage markers in each subpopulation. **j** Scatter plot indicating the correlation of M1- and M2-like index values and the expression of c-Myc. **k** Four categories stratified based on the average of M1- or M2-like index values. Violin plots showing the expression of two epithelial markers (**l**) and two mesenchymal markers (**m**) in categories I, II, III, and IV. *P*-values were determined using a one-way ANOVA followed by Tukey test. Source data for d are provided as a Source Data file.

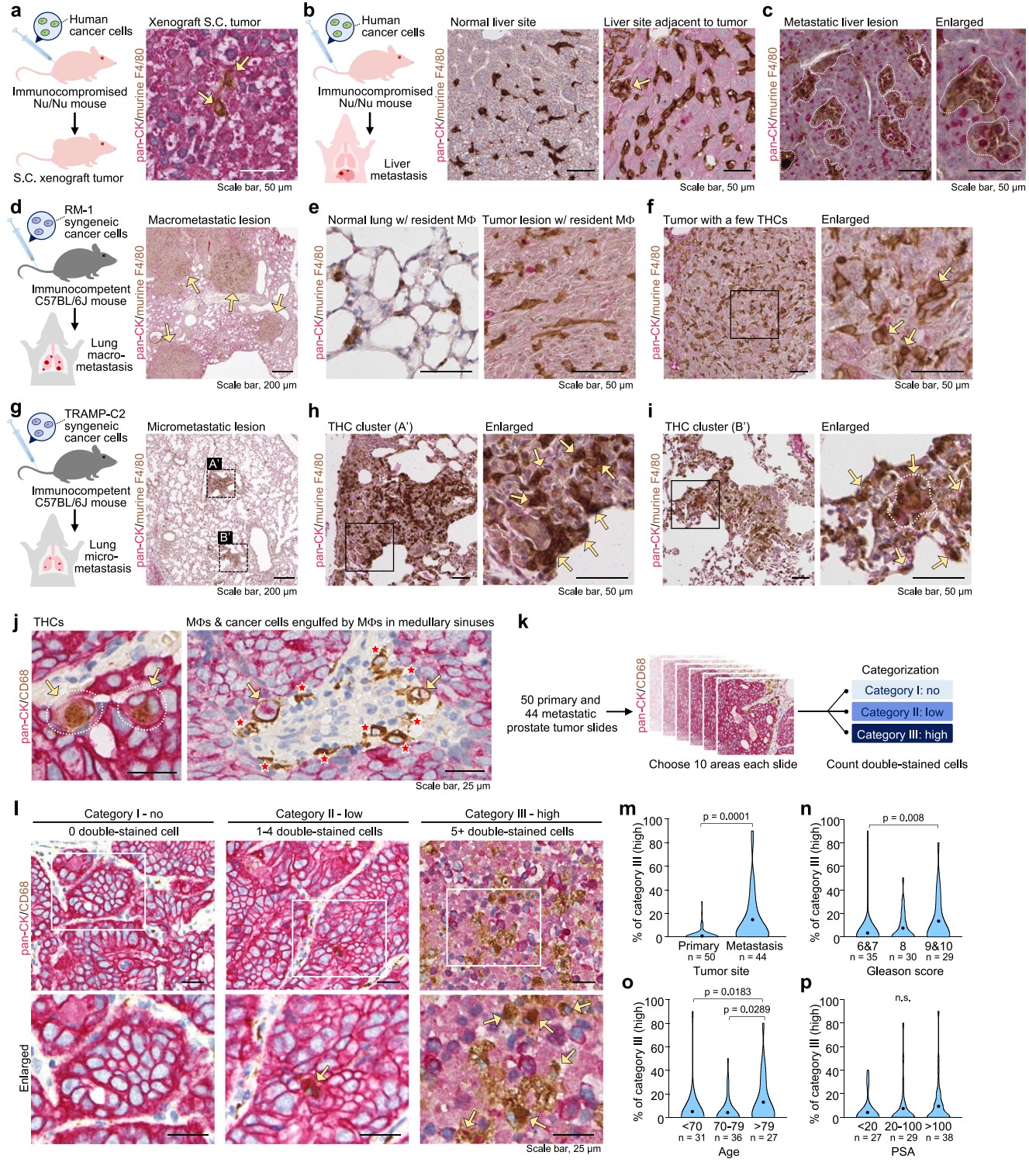

target expression in the hybrid tumor (Fig. 6c). The hybrid tumor negatively expressing neuroendocrine (NE) markers is analogous to double-negative prostate cancer (DNPC), which is immunoevasive and associated with metastasis (Supplementary Fig. 7a)[42]. Co-cultured putative THCs were also less sensitive to AR signaling inhibitor (ARSI) than parental cancer cells, suggesting that these cells underwent transcriptomic changes, shifting from an AR-dependent to AR-independent phenotype (Supplementary Fig. 7b).

To confirm this observation, we conducted gene-signature analyses of four mCRPC cohorts (total $n = 634$, Supplementary Fig. 7c)[43–47].

Bulk RNA-seq was performed on tumor samples of these patients considered for ARSI treatments or standard of care. An empirical Bayes method was used to adjust for data variability among these cohorts due to batch effects[48]. Hierarchical clustering of the data identified four mCRPC groups—two adenocarcinoma-enriched (Adeno-classic and Adeno-immune) and two NE-enriched, as described previously[45]. The Adeno-classic group was known to have high AR activities, while the Adeno-immune group expressed elevated levels of common immune genesets[43]. When additionally including M1- and M2-like macrophage gene profiles mentioned earlier, we divided the Adeno-

**Fig. 5 | Naturally occurring THCs are observed in murine xenograft models and human prostate cancer patients.** C4-2 cells were inoculated into immunocompromised Nu/Nu mice through subcutaneous (**a**) or tail vein (**b, c**) injection (*n* = 2 mice/group). Representative immunohistochemistry (IHC) images of pan-cytokeratin (pan-CK) and murine macrophage marker F4/80 in the subcutaneous tumor (**a**), non-tumor liver sites (**b**), or liver with metastatic tumor (**c**). Increased resident macrophages were observed in regions adjacent to metastatic liver tumor (**b**). Note that clonal expansion of THCs was found in the metastatic liver but not in the subcutaneous tumor. Scale bar, 50 μm. **d–i** Murine syngeneic prostate RM-1 (**d–f**, *n* = 3 mice) or TRAMP-C2 (**g–l**, *n* = 2 mice) cancer cells were inoculated into immunocompetent C57BL/6J mice through tail vein injection. Representative IHC images of pan-CK and F4/80 identified macrometastatic (d-arrows) and micro-metastatic (**g**-A′ and B′) lesions in the lungs of RM-1 and TRAMP-C2 inoculated mice, respectively. Infiltrated macrophages (**e**) and a few THCs (f-arrows) were observed in the RM-1 macrometastatic lesions, while more infiltrated macrophages and more

THCs (**h, i**-arrows) were found in the TRAMP-C2 micrometastatic lesions. These lesions circled by dashed lines A′ and B′ were enlarged in h and i to show the THC clusters, respectively. **j–p** IHC analysis of primary prostate tumors (*n* = 50 specimens) and metastatic tumors (*n* = 44 specimens). Representative images of tissue sections showing THCs (expressing pan-CK and CD68, arrows), macrophages (expressing CD68 only, asterisks), and cells engulfed by macrophages. A bi-nucleated THC was shown (**j**). Workflow of THC analysis in human prostate tumors. Ten regions of each section were categorized into no-, low- (1–4), or high- (>5) THCs (**k**). Representative image of each Category (**l**). More Category III regions were identified in metastatic tumors (**m**). Patients with higher Gleason scores (**n**), older age (**o**), or higher PSA level (**p**) had more Category III regions in the tumor sections. *P*-values were determined using a two-sided unpaired *t*-test (**m**) and a one-way ANOVA followed by Kruskal–Wallis test (**n–p**). Source data for **m–p** are provided as a Source Data file.

immune group into subtypes I and II (Fig. 6d). In addition to having higher M1- and M2-like gene expression, subtype I had even greater levels of the aforementioned immune genesets than subtype II or the Adeno-classic group (Fig. 6e). As expected, there was a trend that both Adeno-immune subtypes expressed lower levels of *AR*, *FOXA1*, or AR targets than the Adeno-classic group (Supplementary Fig. 7d). Moreover, we found that Adeno-immune subtype I tended to have even lower AR signaling activities (i.e., *FOXA1* and *TMPRSS2*) than subtype II, suggesting the former having a diminished benefit from ARSI treatments. Since THCs express the same immune gene signatures of Adeno-immune subtype I, we suggest that these cells contribute to the development of mCRPC, particularly for DNPC.

### THCs exhibit epithelial-mesenchymal plasticity for traveling adaptation in the vasculature

While circulating tumor cells (CTCs) shed from primary or secondary sites contribute to increased metastasis[49], we sought to determine whether THCs are similarly discharged into the bloodstream for distant dissemination. Therefore, CyTOF was used to assess how frequently CTCs and THCs were present in PBMCs of mice carrying subcutaneous human prostate tumors tagged with EGFP (Fig. 7a, Supplementary Table 6). The main compositions of PBMCs were murine CD45[+]/EGFP[-] cells (45.8%), CD45[+]/Gr-1[+]/EGFP[-] myeloid-derived suppressor cells (MDSCs; 44.4%), and CD45[+]/F4/80[+]/EGFP[-] macrophages (5.4%) (Fig. 7b). CTCs (0.02%) were identified based on the expression of human EGFP and EpCAM, but negative for murine markers (Fig. 7c, Supplementary Fig. 8). Circulating THCs (1.6%), identified as CD45[+]/F4/80[+]/EGFP[+]/EpCAM[+] discharged ~80X more than CTCs in the murine blood.

Circulating THCs had higher expression levels of the aforementioned mesenchymal markers than CTCs, while both cell types expressed similar levels of epithelial markers (Fig. 7d, e). These THCs had higher expression levels of eight M1- and M2-like macrophage markers than CTCs or murine macrophages (Fig. 7f, g). PhenoGraph clustering stratified seven THC subpopulations, which were later aligned by increased c-Myc levels (Fig. 7h, i). Like those observed in metastatic THCs, elevated c-Myc was associated with the upregulation of epithelial, mesenchymal, M1- and M2-like markers in subpopulations 1, 2, and 7 of circulating THCs. Concordantly high expression of these polarization markers was also linked to the upregulation of epithelial and mesenchymal markers in hybrid cells Category IV (Fig. 7j–l). Our observation suggests that circulating THCs have enhanced epithelial-mesenchymal plasticity[50], allowing them to cope with vascular stress.

To test how THCs endure hemodynamic stress, we conducted a simulation experiment in which THCs and parental macrophages and cancer cells were subject to a shear stress test of 7.2 dyn/cm² in ibidi microchannels, similar to the condition in human venule (Supplementary Fig. 9a)[51]. CyTOF profiling grouped 17,742 cells without and with the stress test into 20 subpopulations (Supplementary Fig. 9b).

THCs were identified based on the co-expression of CD45 and EpCAM markers (Supplementary Fig. 9c). We found increased expression of CK19, but not EpCAM, in THCs, parental cancer cells, and parental macrophages after the circulation (Supplementary Fig. 9d). However, a similar increase in two mesenchymal makers was observed in THCs and parental macrophages, but not in parental cancer cells (Supplementary Fig. 9e). When further aligning these subpopulations by increasing Vimentin expression, we observed that THCs and macrophages underwent a major shift to subpopulations with high mesenchymal expression after the circulation (Supplementary Fig. 9f). This increase of mesenchymal proteins likely supported a flexible cellular structure, allowing circulating THCs and macrophages to survive in the circulation[52,53]. In contrast, circulating cancer cells fall short of this flexibility, resulting in decreased viability after the stress test (Supplementary Fig. 9g, h).

### THCs are more prevalent than CTCs in blood samples of prostate cancer patients

To further show that THCs could disseminate into the blood of prostate cancer patients, we conducted immunofluorescence analysis of circulating cells retained on a microfilter with 7 μm pores[52]. These cells co-expressed epithelial EpCAM and macrophage CD86, some of which had large sizes (70-200 μm) known as giant macrophage-like cells (Fig. 8a)[54]. We then used CyTOF to determine the frequency of THCs in PBMCs of 16 patients. CyTOF profiling of ~2.9 million PBMCs identified ten main populations based on differential expression patterns of different myeloid and epithelial markers (Fig. 8b, c, Supplementary Fig. 9a, b, Supplementary Table 7). Two minor populations—CTCs (CD45[-]/CD3[-]/CD19[-]/CD56[-]/CD66b[-]/EpCAM[+], 0.04%) and circulating THCs (CD45[+]/CD3[-]/CD19[-]/CD56[-]/CD14[+]/EpCAM[+], 0.2%) were identified, with the latter being ~5 times more abundant than the former. CTCs appeared to express higher levels of two additional epithelial markers, CK19 and MUC-1, and two AR targets, PSMA and PSA, than those of circulating THCs (Supplementary Fig. 9b, c). Consistent with the findings of murine PBMCs, THCs expressed slightly higher levels of macrophage polarization and mesenchymal markers than monocytes/macrophages but had substantially elevated expression of these markers than CTCs (Fig. 8d, e). When further categorizing THCs and monocytes/macrophages into different subpopulations based on the median expression levels of CD86 and CD163, we found that a higher proportion (38%) of THCs had concordant expression of these markers (i.e., dual M1- and M2-like phenotype) than monocytes/macrophages (Fig. 8f). Increased expression of Vimentin, Fibronectin, and CK19 in THCs was also associated with this dual phenotype, confirming our observation in the murine PBMCs (Fig. 8g, h). Then, we compared the ratios of THCs to CTCs in patients with BCR-only *versus* those later having castration resistance and/or metastasis (CR/MET) (Supplementary Table 7). Despite the limited sample size, higher ratios were

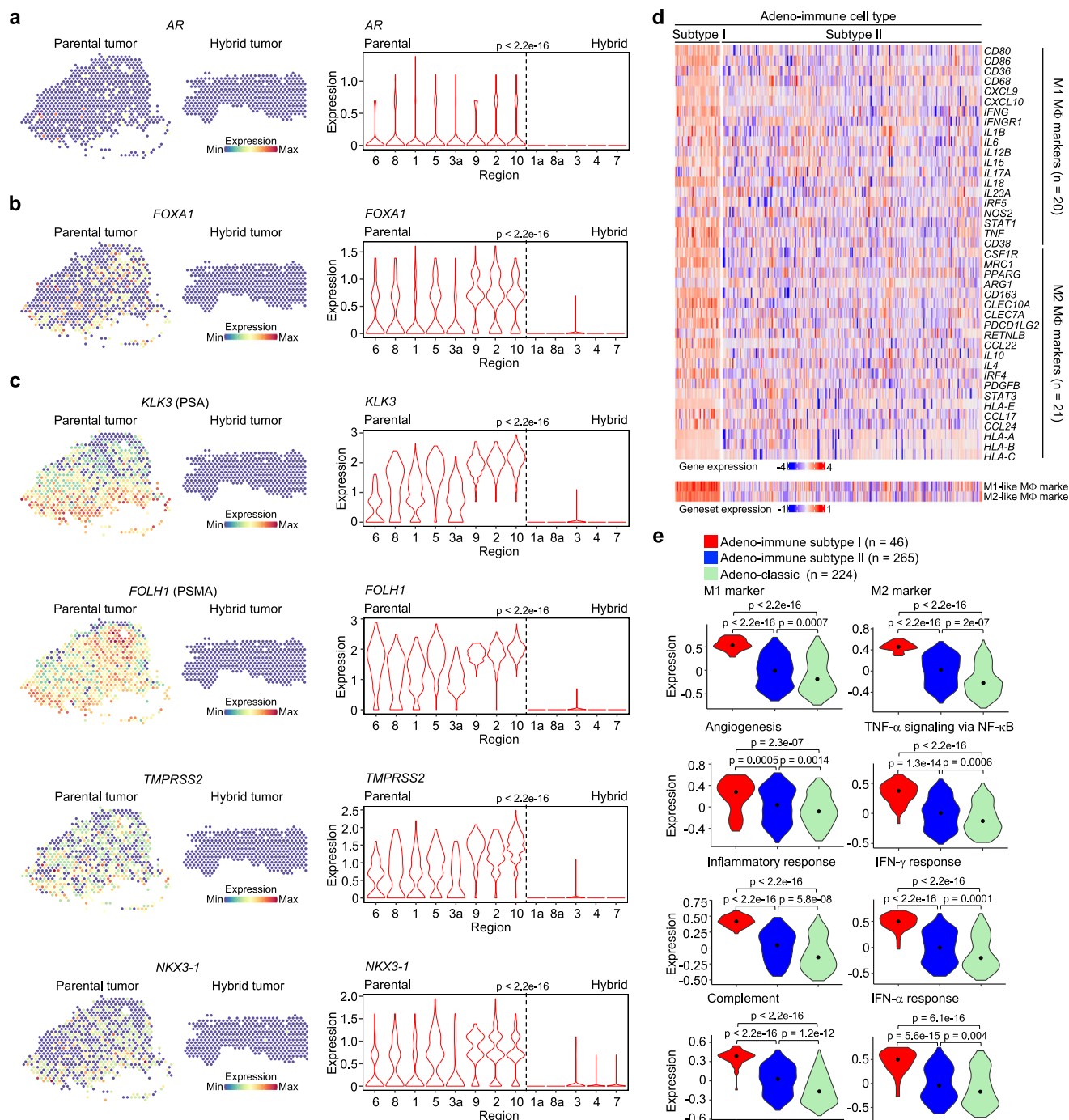

**Fig. 6 | THCs resemble a subtype of metastatic castration-resistant prostate cancer (mCRPC). a–c** Spatial transcriptomics analysis of the androgen receptor (AR) signaling in the parental and hybrid tumors. Left: representative spatial heatmap showing the expression of human *AR* (**a**), *FOXA1* (**b**), or the AR-target genes *KLK3*, *FOLH1*, *TMPRSS2*, and *NKX3-1* (**c**) in parental tumor regions 1, 5, and 6, and hybrid tumor regions 4 and 7. Right: violin plots showing the mean enrichment scores of the aforementioned genes in each region. Hybrid tumor displayed very low expression of AR-target genes. **d, e** Two subtypes were identified from the

Adeno-immune group of mCRPC from four published RNA-seq datasets based on M1- and M2-like macrophage gene expression[43–47]. Heatmap depicting elevated expression of M1- and M2-like macrophage genes in Adeno-immune subtype I (**d**). Violin plots showing the expression of eight immune-related genesets in Adeno-classic, Adeno-immune subtype I, and Adeno-immune subtype II cancer cells (**e**). See also Supplementary Fig. 7c, d and Supplementary Table 1. *P*-values were determined using a two-sided Wilcoxon rank-sum test (**a–c**) and a two-sided unpaired *t*-test (**e**).

found in CR/MET patients, suggesting the use of THCs as a putative biomarker to predict the outcome of advanced cancers (Fig. 8i, j).

## Discussion

Hybrid cells are the progeny of cellular fusion involved in development and differentiation processes[55]. The event also occurs during tumorigenesis, generating hybrid cells often thought to be more malignant

than parental tumor cells in colorectal, pancreatic, lung, and renal cancers and melanoma[22,23,33,56–60]. Aichel first proposed the theory of cancer cell fusion that tumor cells fusing with macrophages or other leukocytes display aggressive abilities for distant metastasis over a century ago[61]. This heterotypic fusion generates a hybrid cell by assimilating two nuclei into a synkaryon or keeping two or more nuclei as a heterokaryon[59]. Recently, two populations of hybrid cells were

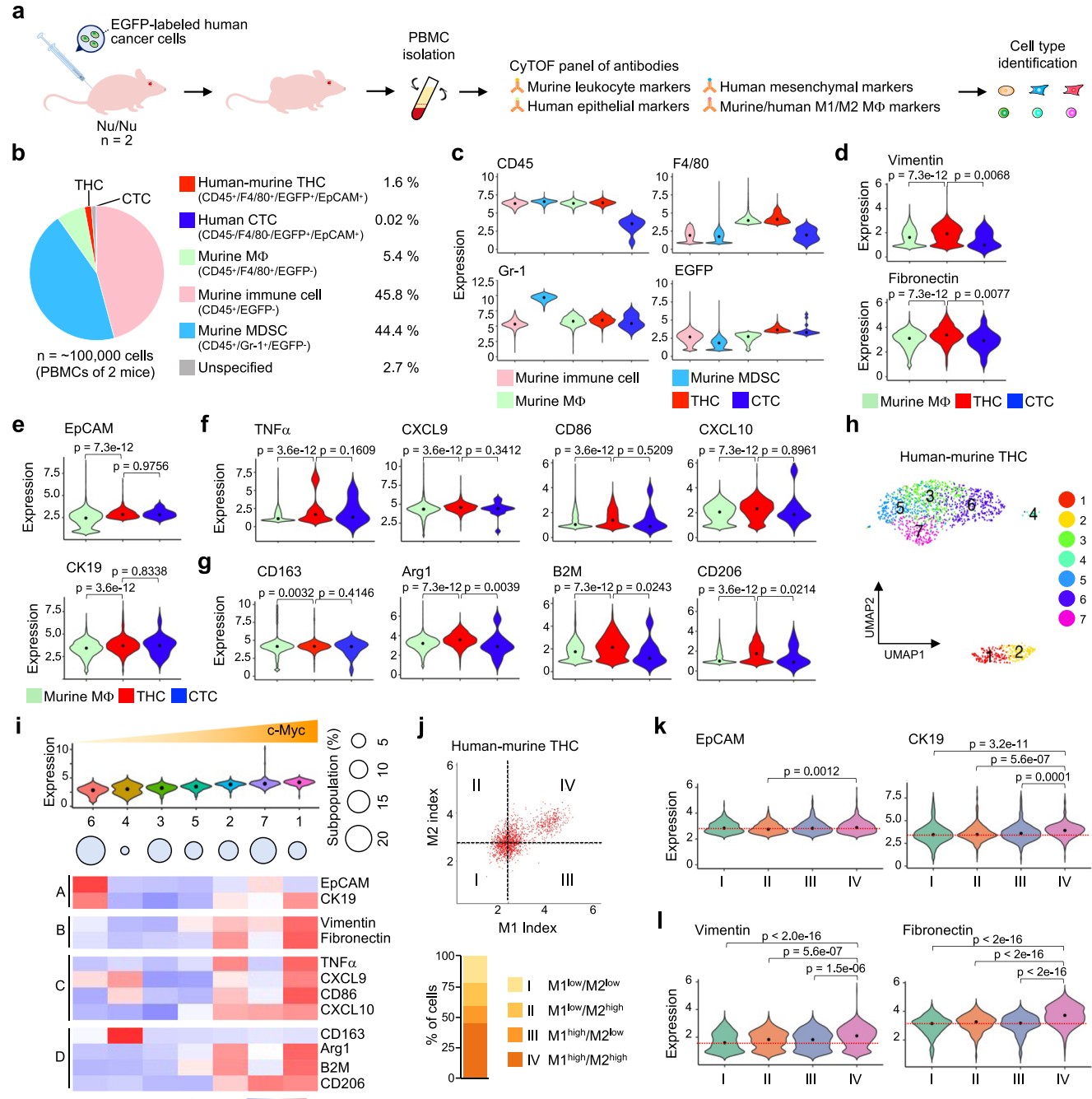

**Fig. 7 | Naturally occurring THCs gain more mesenchymal features in the vasculature. a** Schematic diagram of characterizing THCs in the blood circulation of xenograft mice. Nu/Nu mice were subcutaneously inoculated with EGFP-labeled C4-2 cancer cells. The PBMCs isolated from xenograft mice were analyzed by CyTOF. See also Supplementary Table 3. **b** Pie chart showing the proportions of human-murine THCs, human CTCs, murine macrophages, murine myeloid-derived suppressor cells (MDSCs), other murine immune cells, and unspecified cells in murine PBMCs. **c** The expression of murine CD45, F4/80, Gr-1, and EGFP in the five cell types. **d**–**g** The expression of mesenchymal (**d**), epithelial (**e**), M1- (**f**), and M2-like (**g**) macrophage markers in murine macrophages, human-murine THCs, and

CTCs. **h**, **i** The human-murine THCs were further stratified into seven subpopulations by PhenoGraph clustering (**h**) and aligned by increasing expression of c-Myc (**i**). Heatmap showing the expression of epithelial (A), mesenchymal (B), M1- (C), and M2-like (D) macrophage markers in these subpopulations. **j** Upper: scatter plot showing the M1- and M2-like index values of circulating human-murine THCs in murine PBMCs. Lower: cells were grouped into four categories based on the average of M1- or M2-like index values. **k**, **l** Expression of epithelial (**k**) and mesenchymal (**l**) markers in the four categories shown in (**j**). *P*-values were determined using a one-way ANOVA followed by Tukey test.

characterized—circulating hybrid cells (CHCs) and cancer-associated macrophage-like cells (CAMLs)[60]. The former can result from spontaneous fusion, while the latter is considered a macrophage masquerader containing incompletely digested tumor materials[60,62]. Our present study suggests that most THCs generated in co-culture fit the description of CAMLs. Frist supporting evidence comes from our

karyotypic analysis of metaphase spreads that THCs had hyperdiploid chromosome numbers slightly larger than that of parental macrophages (Supplementary Fig. 4d). If THCs were derived from spontaneous fusions, we assume that their nuclei have polyploid chromosomes acquired from both parental cells. Second, our CD47-SIRPα study implies that weak "don't-eat-me" signals can facilitate the

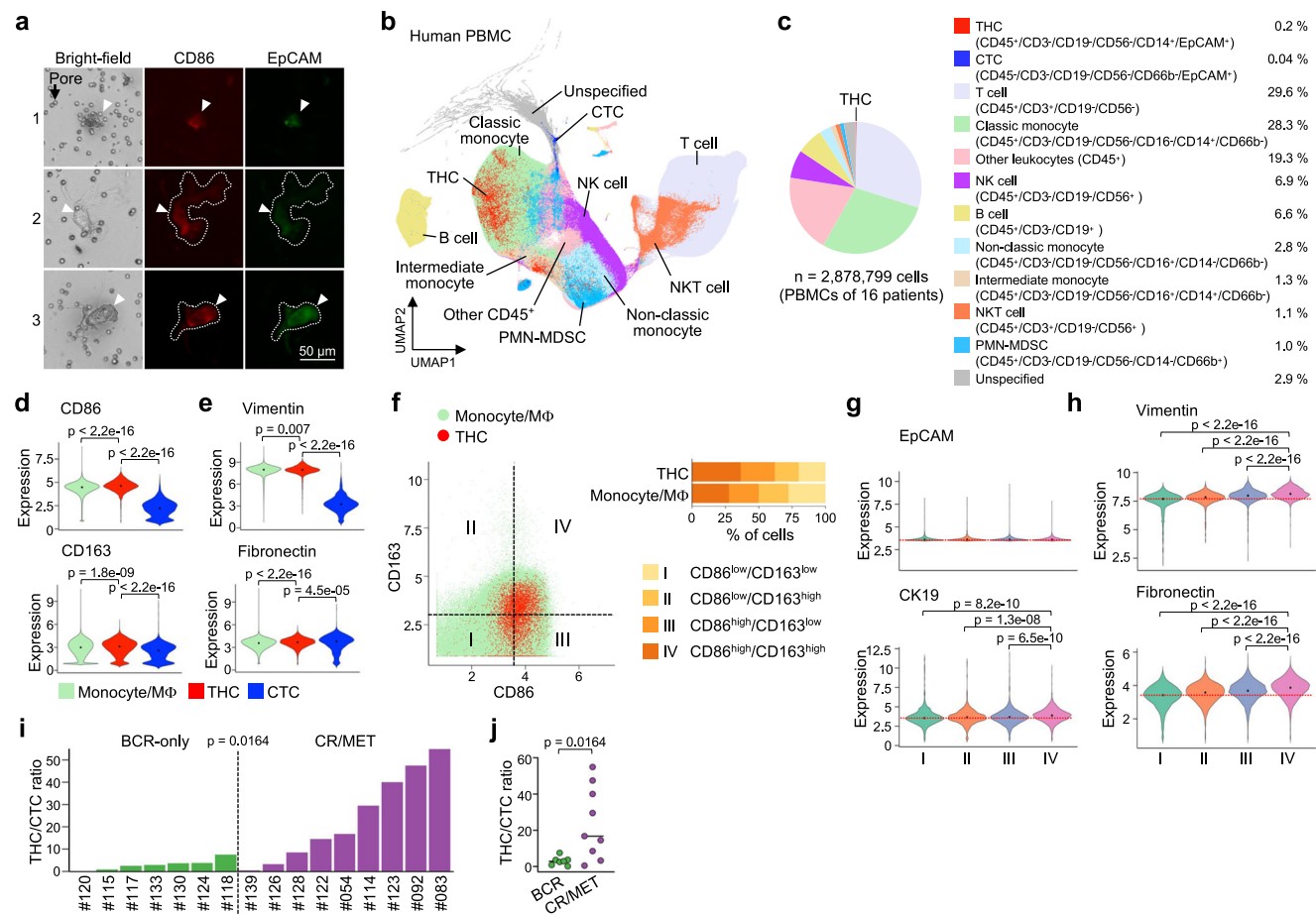

**Fig. 8 | THCs are present in PBMCs of prostate cancer patients.**
**a** Immunofluorescence images of cells retained after filtering out small blood cells through the ScreenCell® pores. Arrowhead, THCs expressed both macrophage CD86 and epithelial EpCAM markers. Dashed lines, giant macrophage-like THCs. These images are representative of >100 putative THCs analyzed from 10 patients. Scale bar, 50 μm. **b**–**j** PBMCs of prostate cancer patients were analyzed by CyTOF (*n* = 16 patients). **b** UMAP depicting different cell types based on their specific markers. CTC, circulating tumor cell; NK, natural killer cell; NKT cell, natural killer T cell; PMN-MDSC, polymorphonuclear myeloid-derived suppressor cell. See also Supplementary Fig. 9 and Supplementary Table 6. **c** Pie chart showing the proportions of each cell type identified by CyTOF. **d**, **e** The expression levels of

mesenchymal markers Vimentin and Fibronectin (**f**) and monocyte/macrophage-related markers CD86, CD163, CD14, and CD16 (**g**) in monocytes/macrophages, THCs, and CTCs. **f** Left: scatter plot showing the expression levels of CD86 and CD163 in monocytes/macrophages and THCs. Cells were divided into four categories based on the median expression of CD86 and CD163. *Right*: proportion of the four categories in THCs and monocytes/macrophages. **g**, **h** The expression levels of epithelial (**g**) or mesenchymal (**h**) markers of putative THCs in the four categories. **i**, **j** Ratios of THC numbers to CTC numbers in individual patients. BCR biochemical recurrence, CR/MET castration resistance and/or metastasis. *P*-values were determined using a one-way ANOVA followed by Tukey test (**d**, **e**, **g**, **h**) and two-sided unpaired *t*-test (**i**, **j**). Source data for **i**, **j** are provided as a Source Data file.

formation of CAMLs through incomplete phagocytosis. This fusion phenomenon is analogous to Stockholm syndrome, where the victim (cancer cell) develops a symbiotic bond with the captor (macrophage) for survival. While the present study is limited to one "don't-eat-me" signaling pathway, further research may explore if THC formation occurs through inadequate levels of other anti-phagocytic ligands in tumor cells[7–10]. Antibodies against these "don't-eat-me" signals have now been used as macrophage checkpoint inhibitors to restore the phagocytic ability of macrophages[5]. The unintended consequence is that partial phagocytosis may take place, giving rise to malignant THCs. Therefore, our present study highlights a potentially overlooked effect of immunotherapy, presenting opportunities to investigate agents targeting the Achilles' heel of THCs.

THCs newly formed in co-culture probably contained diverse and unstable fusogenomes, and therefore only a small fraction of viable THCs were selected for propagation as tumor xenografts. As a result, we observed 30% of visible tumor growth in mice transplanted with co-culturally enriched THCs. Although purified THCs were not used for xenotransplantation, parental cancer cells or macrophages remaining in co-culture mixtures could be used as internal controls for growth

comparisons. Indeed, our spatial gene profiling revealed that 98.5% of hybrid tumor regions harbored THCs, while the remaining 1.5% (regions 1a and 8a) contained parental cancer cells (Supplementary Fig. 2e). Parental macrophages were not detectable in these regions. The finding suggests that THCs were outcompeting parental cancer cells with a growth advantage in the host microenvironment. These cells, disseminated from xenograft tumors through blood circulation, could later form metastatic lesions in distant organs. To improve the current condition for future work, we can further purify THCs based on flow cytometry sorting for orthotopic inoculation in humanized mouse models, which will be better suited for assessing the growth and metastatic potential of THCs.

Our additional studies of murine models and clinical samples confirmed that hybrid cells occurred de novo in primary prostate tumors, albeit with low frequencies. Whether this minor cell population could contribute to distant metastasis remains to be determined. Nevertheless, we observed THCs being formed more frequently in distant organs through invading CTCs, such as circulating C4-2 or TRMAP-C2 cells in murine models. These low CD47-expressing CTCs could be partially engulfed by resident macrophages, generating THCs

in metastatic organs. To evade the host immunosurveillance and elimination, THCs may exploit c-Myc derived from parental CTCs to upregulate M1- and M2-like macrophage polarization genes to imitate resident macrophages. Overexpressed c-Myc, known to antagonize the AR cistrome in prostate cancer[40], may also promote the transition of THCs from an AR-dependent to an AR-independent phenotype. Therefore, these cells are poised to be selected among AR-dependent parental cells for survival after ARSI treatments, resulting in the development of Adeno-immune subtype I. Unlike the prevailing view of castration resistance attributed to signal-switch mechanisms[63,64], our present finding indicates that THCs are an unexpected player in mCRPC. Since c-Myc contributes to immunoevasive and castration-resistant characteristics of THCs, its therapeutic blocking may be considered an additional treatment to re-sensitize this patient sub-group treated with ARSIs. Moreover, enumeration and the associated molecular profiling of circulating THCs can be performed by CyTOF to monitor treatment responses. Although the sample size of our present murine and human studies is limited, this CyTOF-based liquid biopsy can be useful in determining the prognostic value of THCs in large patient cohorts in the future.

## Methods

### Mice

This study investigated THCs in prostate cancer using only male mice. Nu/Nu (strain 007850) and C57BL/6 J (strain 000664) mice were purchased from the Jackson Laboratory. The experiments were conducted following the animal research guidelines from NIH and approved by the Institutional Animal Care and Use Committee at the University of Texas Health Science Center at San Antonio (UTHSCSA), protocol number ORCA20200124AR. For subcutaneous injection, ~$3 \times 10^6$ co-cultured cells, containing ~50% putative THCs, were mixed with Matrigel (Corning, 354234) and separately inoculated into ten 6-week-old male Nu/Nu mice. For control experiments, U937 macrophages and C4-2 cancer cells were similarly inoculated into four and five mice, respectively. For tail vein injection, ~$5 \times 10^5$ C4-2 cells were separately injected into the dorsal caudal veins of two 6-week-old Nu/Nu mice. Tail vein injections were also performed using mouse syngeneic cell lines RM-1 and TRMAP-C2 on three and two C57/BL6J mice, respectively. Tumor size and body weight of mice were measured twice a week after four weeks post-injection. While the permitted maximal tumor size is 2 cm³ as per the protocol guideline, the growth rate of hybrid tumors was fast and slightly exceeded the limit before tumor harvest. For serial transplantations, tumors were minced into small pieces. About 50 μL of minced pieces were subcutaneously transplanted into 6- to 10-week-old Nu/Nu recipient mice (total 10).

### Patients

All patients were enrolled in the protocol approved by the Institutional Review Board of UTHSCSA or Ditmanson Medical Foundation Chia-Yi Christian Hospital. Written informed consent was obtained from all prostate cancer patients. Peripheral blood was collected from patients initially diagnosed with biochemical recurrence (BCR), and some patients subsequently developed castration resistance and/or metastasis (CR/MET). Solid tumors were collected from 48 patients with primary prostatic adenocarcinoma and 41 patients with metastatic prostate adenocarcinoma. The clinicopathological information of the sample donors is shown in Supplementary Tables 4, 5, and 7.

### Cells

All cell lines were purchased from the American Type Culture Collection (ATCC), and cell authentication was conducted at the ATCC. C4-2 (CRL-3314), 22Rv1 (CRL-2505), DU145 (HTB-81), IC-21 (TIB-186), TRAMP-C2 (CRL-2731), and RM-1 (CRL-3310) cells were maintained in RPMI-1640 (Gibco) supplemented with 10% FBS (Sigma-Aldrich) and 1% penicillin–streptomycin (Gibco). U937 (CRL-1593.2) cells were

maintained in the same manner but with 10% heat-inactivated FBS instead. Human peripheral blood CD14⁺ monocytes (STEMCELL Technologies, 70035) and human peripheral blood mononuclear cells (PBMCs) (ATCC, PCS-800-081) were purchased commercially and certified by the manufacturer. CWC cell lines, derived from hybrid tumors after serial xenotransplantation, were maintained in RPMI-1640 supplemented with 10% FBS and 1% penicillin–streptomycin on ultra-low flaks for long-term culture. All cells were cultured in a humidified incubator at 37 °C with 5% $CO_2$.

To generate stable fluorescence-labeled cells, cells were transduced with lentiviral vectors containing constructs, pLenti-puro3/To/V5-GW/EGFP-Firefly luciferase (Addgene#119816, for C4-2 and 22Rv1), FUGW (Addgene#14883, for DU145), or pEFla-mCherry-P2A-Hydro (Addgene#135003, for U937) for 24 h. Puromycin (4 μg/mL for C4-2 and 22Rv1 cells, Thermo Fisher Scientific) or hygromycin B (200 μg/mL for U937 cells, Thermo Fisher Scientific) was used to select stabilized fluorescent single cells. To generate knockin or knockdown cells, cells were infected with lentivirus containing GFP_CD47_LU plasmid (Addgene#65474, for C4-2) or pcDNA3.2(+)eGFP (Addgene#129020, for C4-2 as scramble control), CD47 shRNA plasmid (Sigma-Aldrich, TRCN0000007836, for 22Rv1 and DU145), MYC shRNA plasmid (Sigma-Aldrich, TRCN0000174055, for CWC), or PLKO.1-Scramble plasmid (Addgene#136035 for 22Rv1, DU145, and CWC as scramble control) for 24 h. The target sequences of shRNA were listed in Supplementary Table 2. Lentiviral vectors were packaged using 3rd Generation Packaging Mix (Applied Biological Materials) following the manufacturer's protocol.

### Macrophage isolation and differentiation

For U937 macrophage differentiation, cells were incubated with the culture media containing 100 ng/mL phorbol 12-myristate 13-acetate (PMA) (Sigma-Aldrich, P8139) on an ultra-low attachment flask for three days. Macrophages were collected by centrifugation and then dissociated into single-cell suspension using trypsin solution (Gibco) at 37 °C for 5 min. For inducing the differentiation of human primary macrophages, human peripheral blood CD14⁺ monocytes (STEMCELL Technologies, 70035) were thawed and cultured in ImmunoCult™-SF macrophage medium (STEMCELL Technologies, 10961) containing 50 ng/mL of human recombinant M-CSF (STEMCELL Technologies, 78057) for four days. Then, the media were refreshed, and cells were further cultured for two more days. The attached macrophages were harvested by incubating in the macrophage detachment solution (PromoCell, C41330) at 37 °C for 15 min.

For murine bone marrow-derived macrophages (BMDM) preparation, bone marrow cells were freshly isolated from the femur of untreated 6−8 weeks old C57BL/6J mice. To prepare BMDM differentiation media, L-929 (ATCC, CCL-1) cells were maintained in DMEM (Gibco) supplemented with 10% FBS (Sigma-Aldrich) and 1% penicillin–streptomycin (Gibco), then the cultured media were collected and filtered through a PVDF 0.45 μm syringe filter (Sigma-Aldrich, SLHVR33RS). The bone marrow cells were then cultured with the L-929 conditional media for 7 days and refreshed the media on day 3 to induce murine macrophage differentiation. The differentiated BMDM were de-attached through trypsin solution at 37 °C for 5 min and collected by gentle scraping for future experiment use.

### Co-culture experiments and flow cytometry

To perform phagocytosis assays, EGFP-labeled human prostate cancer cells or murine prostate cancer cells were labeled with Tag-it Violet™ proliferation and cell tracking dye (TiV) (1:1000 Biolegend, 425101) for 20 min and co-cultured with the macrophages or un-differentiated monocytes at 1:2 ratio on an ultra-low attachment dish. The proportion of THCs was analyzed through flow cytometry. To study the effect of "don't-eat-me" signals in THC formation, EGFP-labeled cancer cells were pre-incubated and blocked with media containing 10 μg/mL of

anti-CD47 antibody (eBioscience, 14-0479-82, clone B6H12), anti-CD24 antibody (Novus Biologicals, NB100-64861, clone SN3), anti-PD-L1 antibody (BioXCell, BE0285, clone 29E.2A3), or the isotype control antibody (BioXCell, BE0083) for 6 h. The cancer cells were then co-cultured with the macrophages for phagocytosis assays. To study the effects of SIRPα in co-culture, U937 macrophages and C4-2 cells were co-cultured and treated with 20 μg/mL SIRPα (R&D Systems, 9378-SA) for 4 days. To enrich the stem-like subpopulation, C4-2 cells were cultured in PRIME-XV Tumorsphere SFM (Fujiflm, 91130) supplemented with heparin (Sigma Aldrich, H3149) and hydrocortisone (Sigma Aldrich, H0135) for 7 days. To study the effects of androgen receptor inhibition on the THCs and the parental cancer cells, U937 macrophages and C4-2 cells were co-cultured for one day to form the THCs, and the co-culture mixture was treated with DMSO or 5, 10, or 20 μM of enzalutamide. To study the T cell activation, PBMCs were thawed and co-cultured with CWC cells at a 10:1 ratio and maintained in ImmunoCult™-XF T cell expansion medium (STEMCELL Technologies, 10981) for 5 days. For T cell activation, the co-culture mixture was treated with 2.5 μg/ml of Lectin (Sigma-Aldrich, L8777) and additionally incubated for 2 days. The co-cultured cell pellets were collected and gently digested into a single-cell suspension using trypsin solution before subsequent experiments.

For flow cytometry analysis, co-cultured single-cell suspensions were stained with Zombie-NIR dye (1:100, Biolegend, 423106) for 20 min to exclude the dead cells. After washing with Cell Staining Buffer (Biolegend, 420201), cells were pre-incubated with Human TruStain Fc-X (1:20, Biolegend, 422302) for 10 min to block non-specific staining, and then incubated with fluorescence-labeled antibodies for 30 min at 4 °C in darkness. The following antibodies were purchased from Biolegend: BCL-2-Alexa Fluor 647 (1:20, clone 100), CD4-PE (1:20, clone RPA-T4), CD8-Alexa Fluor 647 (1:20, clone SK1), CD25-FITC (1:20, clone BC96), CD45-Brilliant Violet 785 (1:20, clone HI30), CD86-PE (1:25, clone IT2.2), and c-MYC-Alexa Fluor 647 (1:20, clone 9E10). F4/80-PE was purchased from Thermo Fisher Scientific (1:20, MA5-16631). Flow cytometry analyses were performed using BD FACSCelesta Cell Analyzer (BD Biosciences), and data were analyzed by FlowJo software (BD Biosciences). The macrophage-engaging phagocytosis was quantified and shown as the percentage of EGFP⁺/CD86⁺ macrophages or monocytes for human co-culture models or TiV⁺/F4/80⁺ for murine co-culture models in Zombie⁻ live-cell population. Representative gating strategies were shown in Supplementary Fig. 10.

### Cell viability and apoptosis assay
For CD47 anti-apoptosis function, cancer cells were treated with 20 ng/mL TNFα (Novus Biologicals, 210-TA/CF) or 20 ng/mL TNFα combined with 20 μg/mL SIRPα (R&D Systems, 9378-SA) for three days. For cell viability assay, 4000 cells were seeded in 96-well culture plates for treatments and lysed using CellTiter-Glo Luminescent Cell Viability Assay Kit (Promega, G7571) at the end of the experiments. The luminescent signals were detected by Thermo Luminoskan Ascent Microplate Reader (Thermo Fisher Scientific). For apoptosis assay, cells were resuspended in Alexa Fluor 647 Annexin V (1:20, Biolegend, 640912) solution and incubated for 15 min at room temperature. Then, the Annexin V binding buffer (Biolegend, 422201) was added to the reaction, and the signals were analyzed by BD FACSCelesta Cell Analyzer (BD Biosciences) and FlowJo software.

### Cell imaging and IncuCyte
EGFP-labeled C4-2 cells and mCherry-labeled U937 macrophages were co-cultured in a six-well plate and incubated in IncuCyte S3 Live-Cell Analysis Instrument (Sartorius). The live-cell images were taken every hour and processed through Incucyte Base Analysis Software (Sartorius). For THC imaging, the THCs were isolated from co-culture mixture of U937 macrophages and C4-2 cells through BD FACSAria Fusion Cell Sorter (BD Biosciences), and cell images were acquired through EVOS

cell imaging system (AMG). For spheroid formation assay, CWC cells were resuspended with Matrigel (Coring, 354234) and incubated with RPMI-1640 (Gibco) supplemented with 10% FBS (Sigma-Aldrich) and 1% penicillin−streptomycin (Gibco) for eight days. For bone-in-culture array (BICA), fresh femur and spine bones were isolated and fragmented into 1−2 mm pieces and then incubated in RPMI-1640 (Gibco) supplemented with 10% FBS (Sigma-Aldrich) and 1% penicillin−streptomycin (Gibco) on an ultra-low 12-well plate for 3 weeks. The live images of enriched THCs in BICA were acquired through EVOS fl Fluorescence Microscope (AMG).

### Western blotting and capillary western immunoassay (WES)
Cell lysates were extracted using RIPA buffer (Thermo Fisher Scientific, 89900) supplemented with Pierce Protease and Phosphatase Inhibitors (Thermo Fisher Scientific, A32961). The protein concentration was measured using a BCA protein assay kit (Thermo Fisher Scientific, 23227). For western blotting, cell lysates were fractionated by 10% SDS-polyacrylamide gels and transferred onto nitrocellulose membranes. The membranes were incubated in Tris-buffered saline containing 5% skimmed milk and 0.1% Tween 20 for 1 h to remove the undesired background. Then the membranes were incubated with TBS containing 0.1% Tween 20 and anti-CD47 antibody (1:1000, eBioscience, 14-0479-82, clone B6H12), anti-GAPDH antibody (1:10000, Cell Signaling Technology, 2118S), or anti-β-Actin (1:5000, R&D Systems, MAB8929) at 4 °C for 16−18 h After incubating with horseradish peroxidase (HRP)-conjugated secondary antibody (Santa Cruz Biotechnology, sc-2357 (1:2000) or sc-2005 (1:1000)), the chemiluminescent signal was detected using Western Lightning Plus ECL (Perkin Elmer, NEL103E001EA) and imaged using G:box imaging system (Syngene). The uncropped images of Western blotting were shown in Supplementary Fig. 11. Quantification of the CD47 expression was performed using Fiji ImageJ software.

For WES, cell lysates were loaded on the Jess/Wes 12−230 kDa pre-filled plates (ProteinSimple, SM-W004), and specific proteins were recognized using the commercial antibodies−anti-BCL-2 (1:25, R&D Systems, MAB827), anti-CD47 (1:25, Abcam, ab284132), anti-c-Myc (1:25, R&D Systems, MAB3696), anti-Lamin A/C (1:50, Cell Signaling Technology, 2023 S), and anti-β-Actin (1:100, R&D Systems, MAB8929). The bound antibodies were visualized and detected using the anti-rabbit or anti-mouse HRP detection module (no dilution, ProteinSimple, DM-001 or DM-002) and quantified by Compass for SW software. The uncropped images of WES were shown in Supplementary Fig. 12.

### Proximity-ligation assay (PLA)
C4-2 cells were seeded in a chamber slide and treated with PBS or 20 ng/mL SIRPα (R&D Systems, 9378-SA) for 1 h. After the removal of free SIRPα, cells were fixed with 4% paraformaldehyde for 10 min and washed twice with PBS containing 0.1% Tween 20 (PBST). The PLA was performed using Duolink In Situ Red Starter Kit (Sigma-Aldrich, DUO92101) following the manufacturer's instructions. In brief, cells were incubated with Duolink blocking solution at 37 °C for 1 h and then incubated with the antibodies of CD47 (1:100, eBioscience, 14-0479-82, clone B6H12) and SIRPα (1:100, Cell Signaling Technology, 13379) at 4 °C for 16−18 h. After washing, the PLUS and MINUS PLA probes were added to the cells and incubated at 37 °C for 1 h. For ligation and amplification, cells were incubated with Duolink ligation solution and then in Duolink amplification solution. After washing, cells were mounted with Duolink PLA mounting medium with DAPI. The images were acquired by LSM 710 confocal microscope (Carl Zeiss) and analyzed by Fiji ImageJ software.

### Cytogenetic analysis
Cytogenetic analysis was performed on metaphase spreads obtained from two CWC cell lines, parental C4-2 cells, and parental U937

macrophages. Culture initiation, maintenance, and harvest were carried out using standard methods. Chromosomes were G-banded using trypsin and analyzed using a CytoVision image analysis system (Applied Imaging, Santa Clara, CA).

## Spatial transcriptomics

One fresh parental C4-2 tumor and one fresh hybrid tumor were embedded in Tissue-Tek optimal cutting temperature (OCT) (Sakura Finetek, 4583) and stored at −80 °C. The tumors were sectioned by CryoStar NX50 Cryostat (Epredia) into 10 μm slides and applied to the Visium Spatial Gene Expression slides whih 4992 spots containing spatial barcodes. (10x Genomics). Tissue sections were then incubated with blocking solution for 5 min and stained with DAPI. The images of tissues were acquired by Cytation 5 (BioTek). Tissue optimization and spatial gene expression profiling using Visium spatial tissue optimization reagent kit, Visium spatial tissue optimization slide kit, Visium spatial gene expression reagent kit, and Library construction kit followed the instructions of 10x Genomics (document numbers CG000238 and CG000239). Briefly, the tumor sections were incubated with the permeabilization enzyme at 37 °C for 30 min, and the cDNA was synthesized by RT Master Mix in a thermal cycler. After KOH incubation and EB buffer washing, a second strand synthesis was performed, and the cDNA was released from the tissue by KOH incubation. Then, the cDNA was amplified by the Amplification Mix in a thermal cycler and cleaned up using SPRIselect reagent (Beckman Coulter). The quality of cDNA was validated with 2100 Bioanalyzer instrument (Agilent). After fragmentation, end repair, and A-tailing, the cDNA was cleaned up again through SPRIselect reagent (Beckman Coulter) and ligated through the Adaptor Ligation Mix to generate the Spatial Gene Expression library. Post-library construction quality control was performed using 2100 Bioanalyzer instrument (Agilent), and the sequencing was conducted by NextSeq 500 with NextSeq 500/550 high output kit (Illumina).

For data analysis, raw sequence files were demultiplexed into library-specific fastq files using mkfastq subcommand in 10X Genomics Space Ranger tool. Human and mouse reference datasets, GRCh38 and mm10, as well as gene models in GTF format from GENCODE v32 and vM23 were downloaded and preprocessed by the reference-building script from 10x Genomics Cell Ranger. The kallisto index command was used to build a GRCh38 and mm10 mixed index[65]. Reads were pseudo-aligned to this genome assembly index by using kb count command with -x 0,0,16:0,16,28:1,0,0. Cell barcodes and unique molecular identifiers were filtered and corrected. BUStools was used to error-correct barcodes based on the whitelist barcodes file from Space Ranger and produce gene-barcode count matrices in Cell Ranger output format[66].

A customized script based on Seurat was used to load and analyze the data. Sctransform, based on the regularized negative binomial models of gene expression, was used to normalize the data and detect high-variance features. Principal component analysis (PCA) was run using the previously determined high-variance features for linear dimensionality reduction, and the first 30 principal components were kept for clustering spots with similar expression profiles based on their projection into a dimensionality-reduced PCA subspace. The Louvain algorithm was used to cluster and t-SNE was used to visualize the high-dimensional data.

Differentially expressed genes in each cluster were identified using the FindAllMarker function built within the Seurat package, and a corresponding p-value was given by the Wilcoxon's test followed by a Bonferroni correction. The significant genes were defined as at least 25% of spots in either of the two clusters have the genes expressed and the expression difference was at least 0.25 on a natural log scale. AUCell was used to calculate any geneset enriched among the expressed genes for each spatial spot[67]. The Hallmark genesets were downloaded from the MSigDB database (http://www.gsea-msigdb.org/gsea/msigdb/index.jsp)[27]. Genesets were also curated based on the literature for major immune cells in both human and mouse, including CD4+ T cells, CD8+ T cells, T reg cells, B cells, M1- and M2-like macrophages, myeloid-derived suppressor cells (MDSCs), natural killer (NK) cells, and fibroblasts. The gene signature scores for all genesets and spatial spots were conducted using the CreateAssayObject command.

## Immunofluorescence staining and immunoFISH

For immunofluorescence staining, cells were fixed with 4% paraformaldehyde (Santa Cruz Biotechnology, sc-281692) at room temperature for 10 min. Tumors were embedded by OCT (Sakura Finetek, 4583) and sectioned into 10 μm slides by CryoStar NX50 Cryostat (Epredia). After washing with PBST, cells were incubated with 10% goat serum (Abcam, ab4781) for 30 min and then incubated with primary antibodies at 4 °C for 16–18 h. After washing with PBST, cells were incubated with secondary antibodies at room temperature for 1 h. Specific cellular proteins were labeled using antibodies and secondary antibodies commercially purchased and used in dilution (v:v) · CD14 (1:100, Abcam, ab183322), CD68 (1:100, Cell Signaling Technology, CST76437), CD86 (1:100, Cell Signaling Technology, CST91882), EpCAM (1:200, Abcam, ab71916 for single staining), EpCAM (1:200, Cell Signaling Technology, CST2929 for double staining), Cytokeratin 18 (1:100, Biolegend, 628404), F4/80 (1:100, Thermo Fisher Scientific, MA5-16363), pan-Cytokeratin (1:200, Cell Signaling Technology, CST4545), Vimentin (1:200, Novus Biologicals, NBP192687), goat anti-mouse IgG Alexa Fluor 594 or Alexa Fluor 647 (1:200, Abcam, ab150116 or ab150115), goat anti-rabbit IgG Alexa Fluor 488 or Alexa Fluor 647 (1:200, Abcam, ab150077 or ab150087), and goat anti-rat IgG Alexa Fluor 594 (1:200, Abcam, ab150168). For THC isolation and identification, -10 ml of patients' blood was mixed with ScreenCell® LC dilution buffer and filtered through ScreenCell® filter (ScreenCell, CC-3LC) following the manufacturer's instructions. After PBS wash, the live cells were stained with anti-EpCAM-FITC (1:100, STEMCELL Technologies, 60136Fl) and anti-CD86-PE (1:100, Biolegend, clone IT2.2) antibodies and the images were acquired through EVOS fl Fluorescence Microscope (AMG).

For immunoFISH, immunofluorescence staining was conducted on cells with a c-Myc antibody (1:100, R&D Systems, MAB3696) and anti-mouse Alexa Fluor 647 (1:200, Abcam, ab150115) as described above. Then the cells were permeabilized by serial incubations of PBS containing 1% Triton X-100, and 0.1 N HCl each for 30 min. Cells were then incubated with 2× SSC solution for 5 min\ and 50% formamide/2X SSC solution (Sigma-Aldrich, F9037) for 1 h. Then, the Gold-dUTP labeled *MYC* FISH probe (1:20, Empire Genomics, RP11-440N18) was added to the cells and denatureation was performed at 76 °C for 3 min. After incubating at 37 °C for 16–18 h in a humidity chamber, cells were subject to a serial of washes with a solution containing 0.4% SSC and 0.3% NP-40, a solution containing 2× SSC and 0.1% NP-40, and PBS for 5 min at room temperature except the first wash was performed at 72 °C. Cell nuclei were stained with DAPI and images were acquired by LSM 710 confocal microscope (Carl Zeiss) with 1 μm z-section in each field. The images were projected and combined using Fiji ImageJ software.

## Immunohistochemistry (IHC)

Paraffin-embedded murine tumors were sectioned into 4-μm slices and sequentially immersed in xylene, 100% ethanol, 95% ethanol, and dH$_2$O twice, each 10 min. Antigen retrieval was performed with Citrate Antigen Retrieval Buffer (Abcam, ab93678, pH 6.0) boiled for 10 min and cooled down for 30 min. Samples were washed with dH$_2$O three times and incubated with a Hydrogen Peroxide Blocking Reagent (Abcam, ab64218) for 10 min. After washing with dH$_2$O and PBST, samples were blocked with 10% goat serum (Abcam) for 1 h and incubated with anti-F4/80 (1:600, Cell Signaling Technology, CST70076) and anti-pan-Cytokeratin (1:100, Cell Signaling Technology, CST4545)

antibodies at 4 °C for 16–18 h. Secondary antibody incubation and chromogenic reaction were performed with SignalStain® IHC Dual Staining Kit (AP, Rabbit, Red/HRP, Mouse, Brown) (no dilution, Cell Signaling Technology, 36084). Briefly, samples were washed with PBST three times and incubated with AP- or HRP-conjugated secondary antibodies at room temperature for 30 min. Protein colocalization was performed by Alkaline phosphate or DAB substrate reagents, and the samples were counterstained with hematoxylin (Vector Laboratories, H-3401). Samples were then sequentially dehydrated by being immersed in 95% ethanol, 100% ethanol, and xylene for 10 seconds each. The images were acquired by slide scanning microscope (Keyence, BZ-X800).

Paraffin-embedded human tumors were sectioned into 4-μm slices and performed pan-cytokeratin (1:300, Sakura Finetek USA, 60-0022) and CD68 (1:400, Zeta Corporation, 50-221-5864) staining through BOND-III automated IHC staining system (Leica Biosystems Newcastle Ltd, 22.2201) following the manufacturer's guidelines. Bond Polymer Refine Red Detection-AP red (Leica Biosystems Newcastle Ltd, DS9390) and BOND Polymer Refine Detection-DAB (Leica Biosystems Newcastle Ltd, DS9800) were used for chronogenesis, and the images were acquired by slide scanning microscope (Keyence, BZ-X800). Ten areas of each tumor section were evaluated and categorized based on the number of CD68$^+$/pan-CK$^+$ cells by two individuals blinded to any clinicopathological information. Areas with no double-stained cells were classified as Category I; areas with 1–4 double-stained cells were classified as Category II; areas with more than five double-stained cells were classified as Category III. Clinicopathological characteristics, tumor staging, therapeutic modalities, and survival data were collected for statistical analysis. All tumors clinically, histopathologically, and immunochemically fulfilled the criteria of prostatic adenocarcinoma after confirmatory reviews. Prostate origins of metastatic tumors were confirmed by NKX3.1-positive expression. A chart review was used to collect clinicopathological data, including gender, age, PSA level, stage, survival time, and illness progression. According to the eighth edition of the American Joint Commission on Cancer (AJCC) staging system, the tumor stage was reevaluated. Two pathologists independently evaluated and confirmed the histopathological characteristics of tumors in the cohort. Clinicopathological parameters of patients were displayed in Supplementary Tables 4 and 5.

### Transwell assay
Three days of co-cultured cells (1.8 × 10$^5$ cells) pretreated with DMSO or 10058-F4 (Sigma-Aldrich, F3680-5MG) were seeded in the lower chamber of the 8 μm 24-well Nunc™ polycarbonate cell culture plate inserts (Thermo Fisher Scientific, 140629) and incubated for 2 h. Then, ~8 × 10$^4$ mouse macrophage IC-21 cells were seeded in the upper chamber and incubated in a humidified incubator at 37 °C with 5% CO$_2$. At the end of the assay, cells on the upper chamber were gently removed with a cotton swab, and the membranes were washed with PBS and fixed with 4% paraformaldehyde. After washing with PBS, the membranes were stained with crystal violet solution for 20 min and washed with dH$_2$O three times. The areas of the migrant cells were quantified using Fiji ImageJ software.

### Chromatin immunoprecipitation-quantitative PCR (ChIP-qPCR) and reverse transcription-qPCR (RT-qPCR)
For ChIP-qPCR, C4-2 cells or U937 macrophages were cross-linked in cultured plates using 1% formaldehyde and quenched by glycine, then washed with cold PBS. Fresh hybrid tumors were minced and placed in a 1.5 mL tube pre-filled with 1 mL cold PBS supplemented with Pierce Protease and Phosphatase Inhibitors (Thermo Fisher Scientific, A32961) per 500 mg. Tissues were mashed with an 18-gauge needle and then with a 21-gauge needle. The triturated tissues were transferred to a 15 mL conical tube, cross-linked with 10 mL PBS containing 1% formaldehyde, and then quenched by glycine. After washing with

cold PBS, the single tumor cells were filtered by a 40 μm cell strainer (Corning, 352340) and collected through centrifugation. The chromatin immunoprecipitation-qPCR (ChIP-qPCR) assays were performed using Pierce Magnetic ChIP Kit following the manufacturer's protocol (Thermo Fisher Scientific, 26157). Briefly, 1 × 10$^7$ cross-linked cells were lysed by Membrane extraction buffer and the nuclei were resuspended and digested by MNase at 37 °C for 15 min. Then the nuclei were resuspended with IP dilution buffer, and DNA sonication was performed by Q800R3 Sonicator (Qsonica) at 20% amplitude with 10 s/on and 20 s/off cycles for 4 min. The size of DNA fragments ranged between 200–1000 bp, while the majority of DNA appeared around 160, 320, and 480 bp. The digested chromatin solutions were incubated with 10 μL of an anti-c-Myc antibody (Thermo Fisher Scientific, PA5-85185), anti-RNA polymerase II antibody (Pierce Magnetic ChIP Kit, Thermo Fisher Scientific, 26157), or control rabbit IgG (Pierce Magnetic ChIP Kit, Thermo Fisher Scientific, 26157) at 4 °C for 16–18 h. Then, the DNA fragments were captured by Protein A/G magnetic beads, washed with IP wash buffers, and then eluted with elution buffer. The proteins were digested by proteinase K at 65 °C for 90 min and the immunoprecipitated DNA fragments were ready for real-time PCR analysis after purifying through a DNA Clean-Up column.

For RT-qPCR, RNAs of cells or fresh tumors were extracted with Direct-zol™ RNA miniprep (Zymo Research, R2052) and the cDNAs were made through a high-capacity cDNA reverse transcription kit (Applied Biosystem™, 4368814) following the manufacturer's instructions. LightCycler 480 SYBR Green I Master (Roche, 04887352001) was used for quantitative real-time PCR, and the percent of input of each sample was calculated. ChIP-qPCR or RT-qPCR primer sequences were listed in Supplementary Table 2.

### Stress test in ibidi microchannels
Co-cultured cells were loaded in a fluidic unit and the hemodynamic stress was performed using an ibidi pump system with μ-slide I Luer 0.6 mm (ibidi, USA, Inc.). The unidirectional flow was controlled by PumpControl Software and the rate was 16.67 mL/min. After circulation, cells were collected through centrifugation for CyTOF analysis.

### Cytometry by time-of-flight (CyTOF) analysis
For cell dissociation, minced tissues of a metastatic murine tumor were incubated with RPMI-1640 containing type V Collagenase (Sigma-Aldrich, C9263, 200 μg/ml), Dispase II (Sigma-Aldrich, D4693, 100 μg/ml), and DNAse I (Roche, 11284932001, 100 μg/ml) at 37 °C for 30 min. After washing with PBS, single cells were filtered through a 40 μm cell strainer (Corning, 352340) and collected by centrifugation. The ACK lysing buffer (Gibco, A10492-01) was used to lyse erythrocytes, and single cells were then washed with PBS. Peripheral blood mononuclear cells (PBMCs) from murine blood were incubated with ACK lysing buffer twice for erythrocyte lysis, and the remaining cells were collected after centrifugation. PBMCs were also isolated from human blood using the Ficoll Paque Plus (Cytiva, 17144002) density gradient centrifugation and lysed with ACK lysing buffer. Cells prepared from the tumor, PBMCs, or ibidi mentioned above were used for subsequent CyTOF analysis.

Heavy-metal isotope conjugated antibodies were purchased from Fluidigm or conjugated in-house, following the manufacturer's instructions. Antibodies used in this study were listed in Supplementary Tables 3 and 6. The staining protocol followed the Maxpar nuclear antigen staining protocol with some modifications. In brief, single-cell suspensions were incubated with Cell-ID Cisplatin (1:10,000, Fluidigm, 201064) at room temperature for 5 min. Cells were then fixed with 3.2% paraformaldehyde (Thermo Fisher Scientific, 28906) and washed with Maxpar Cell Staining Buffer (Fluidigm, 201068). Cells were then incubated with Maxpar Perm-S buffer (Fluidigm, 201066) and TruStain Fc-X (1:20, Biolegend, 422302). The cell surface and cytosol antibodies were pooled and added to the cells for incubation at room

temperature for 1 h. After washing with Maxpar Cell Staining Buffer (Fluidigm, 201068), cell pellets were chilled on ice for 10 min, gently resuspended in cold methanol and incubated on ice for another 10 min. After washing, cells were stained with the pooled nuclear antibodies at room temperature for 30 min and washed by Maxpar Cell Staining Buffer (Fluidigm, 201068), then incubated with Maxpar Fix and Perm Buffer (Fluidigm, 201067) containing 0.125 mM Cell-ID Intercalator-Ir 191/193 (Fluidigm, 201192A) at 4 °C for 16–18 h. Proteomic signals were measured using Helios third-generation mass cytometer (Fluidigm) and normalized by the CyTOF software (Version 6.7.1014, Fluidigm).

CyTOF-generated files underwent signal cleanup for data analysis, and Cisplatin-labeled dead cells were excluded from the analysis using FlowJo software, except for cells derived from a metastatic tumor that were semi-fixed before Cell-ID staining. Single-cell CyTOF data were clustered using PhenoGraph clustering embedded in the cytofkit2 R package[68] and visualized using uniform manifold approximation and projection (UMAP). R package ggplot2 was used for generating the violin plots, scatter plots, heatmaps, and circle plots.

### In silico analysis of single-cell and bulk RNA-seq data

The digital gene-cell matrices of GSE176031 were downloaded from the GEO website[39]. For single-cell RNA-seq analysis, low-quality cells (i.e., less than 500 unique features or 300 transcripts, or a mitochondrial level over 20%) were filtered out. DoubletFinder was also used to remove doublets[69]. As such, 12,761 cells were removed, and 22,084 cells were kept for subsequent analysis. The gene expression matrix was processed and analyzed by Seurat (version 4.0.1)[70]. Merging with and without integration of the samples showed no major difference in the cell type clustering. Therefore, the merged dataset without integration was used for subsequent analysis.

The LogNormalize method was used to determine gene expression levels by normalizing the total read count and multiplying a scale factor of 10,000 for log transformation. The top 2000 most variably expressed genes among the cells were then calculated and returned. The expression of each gene was adjusted with a linear log scale. PCA was run using the previously determined 2000 most variably expressed genes for linear dimensionality reduction, and the first 100 principal components were stored. For graph-based clustering, the first 100 principal components with a resolution of 2 were selected, and the Louvain algorithm was used to optimize the partition of single cells into highly interconnected clusters. For cell type annotation, RCAv2 was used to project single-cell RNA-seq data against the Global Panel reference and compute a correlation matrix indicating the similarity of single-cell transcriptomes to the reference transcriptomes[71]. Then, a clustering-independent annotation approach was used to assign cells to a specific type with high similarity scores. Differentially expressed genes in a cell type were identified using the FindAllMarkers function built within the Seurat package, and a corresponding p-value was given by the Wilcoxon's test followed by a Bonferroni correction. These gene datasets were used as inputs to calculate a gene signature score for each cell type by the AUCell tool[67]. The scoring method in AUCell is ranking-based, independent of each expression unit or the normalization procedure. Since no datasets were available to build a classifier model for THCs, those cancer cells displaying high levels (top 1%) of macrophage gene signatures and those macrophages displaying epithelial gene signatures were selected as potential THCs. These cells had slightly higher UMI (unique molecular identifier) counts or gene numbers than their parental cancer cells or macrophages.

For bulk RNA-seq analysis, batch-effect corrected datasets were obtained from four mCRPC cohorts: Stand Up 2 Cancer/Prostate Cancer Foundation East Coast Dream Team (ECDT), West Coast Dream Team (WCDT), Fred Hutchinson Cancer Research Center (FHCRC), and a small-cell neuroendocrine prostate cancer-enriched cohort from Weill Cornell Medicine (WCM)[43–47]. Gene set variation analysis (GSVA) was performed using the Hallmark genesets in MSigDB combined with our in-house M1- and M2-like macrophage polarization genesets (Supplementary Table 1). RNA-seq data were log2 transformed. gsva method, and Gaussian kernel were used during the non-parametric estimation of the CDF function of expression levels across samples. The magnitude difference between the largest positive and negative random walk deviations was used as the enrichment statistic, which was also denoted as the enrichment score for each gene set. Patients in the Adenoimmune group were further clustered into two subtypes based on the corresponding expression scores of M1- and M2-like genes by GSVA.

### Reporting summary

Further information on research design is available in the Nature Portfolio Reporting Summary linked to this article.

## Data availability

The next-generation sequencing data acquired from spatial transcriptomics have been deposited in the Gene Expression Omnibus (GEO) under accession code GSE194102. The public scRNA-seq datasets used in this study are available in the GEO database under accession code GSE176031; the Microarray datasets Grasso, Taylor, Varambally are available under accession code GSE35988, GSE21034, and GSE3325, respectively. The TCGA PRAD dataset was retrieved from cBioPortal [https://www.cbioportal.org/]. The remaining data are available within the article, Supplementary Information or Source Data file. Source data are provided with this paper.

## Code availability

The code of our customized Seurat script has been placed on GitHub for public access (https://doi.org/10.5281/zenodo.8335859; https://github.com/zhangz3/TMH_NC/).

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

## Acknowledgements

We thank Dr. Shuang G. Zhao at the Department of Human Oncology, University of Wisconsin, Madison, Wisconsin, for providing bulk RNA-seq data for analysis; Dr. Michael W. Y. Chan at the Department of Biomedical Science, National Chung-Cheng University, Chia-Yi, Taiwan, for facilitating the collaborative study of a Taiwanese prostate cancer cohort; and Juan Wang at the Department of Molecular Medicine, UTHSCSA for providing technical help of animal model. The authors acknowledge the assistance of the core facilities at the UTHSCSA: the Genome Sequencing Facility for spatial transcriptomics experiments; the BioAnalytics and Single-Cell Core for CyTOF studies; the Optical Imaging Facility for confocal microscopy; and the Flow Cytometry Shared Resource for cell sorting. This work was supported by the National Institutes of Health grants U54CA217297 and P30CA054174, the Department of Defense grant W81XWH-18-1-051, the Cancer Prevention and Research Institute of Texas grant RP150600, San Antonio Cancer Council, and the Max and Minnie Tomerlin Voelcker Fund to T.H-M.H. J.A.T. is a scholar supported by the National Center for Advancing Translational Sciences, National Institutes of Health grant KL2 TR001118 and a Voelcker Investigator. Z. Liu and K.X. are the Cancer Prevention & Research Institute of Texas scholars.

## Author contributions

T.H.M.H. supervised the study. C.W.C., C.N.H., C.H.L.C., C.L.C. and T.H.M.H. designed the experiments. C.W.C., C.N.H., C.H.L.C., M.C., C.C.C., C.W.H., P.O., M.E.G., L.L.L. and N.A. performed and analyzed the experiments. V.O. and G.V.N.V. performed cytogenetic analysis. C.N.H., Z.Z. and Z. Lai. performed and analyzed spatial transcriptomics experiments. C.W.C., C.N.H., M.C. and C.M.W. conducted CyTOF experiments. M.A.L., C.C.C, C.N.H. and M.C. collected and processed human samples. M.C., C.W.C., C.N.H. and C.L.C. conducted mouse studies. C.N.H., Z.Z., C.H.L.C, X.T. and M.S. performed computational analysis. T.H.M.H., C.W.C., C.H.L.C., C.N.H., Z.Z. and C.L.C. wrote the manuscript., T.H.M.H., C.W.C., C.N.H., C.H.L.C., X.T., N.B.K., K.X., Z. Liu., J.A.T., C.L.C. and A.P.K. edited and revised the manuscript. All co-authors contributed to interpreting and commenting on the manuscript.

## Competing interests

The authors declare no competing interests.
