## [Peer Review File · Nature Communications]

Phagocytosis-initiated tumor hybrid cells acquire a c-Myc-mediated quasi-polarization state for immunoevasion and distant disseminationREVIEWER COMMENTS

Reviewer #1 (Remarks to the Author): with expertise in cancer immunology, macrophages, CD47

In this exciting study by Chou et al., the authors discovered “hybrid tumor cells,” which possess characteristics of both cancer cells and macrophages. The authors showed that these cells derived from macrophage phagocytosis of cancer cells, and CD47-low cancer cells tend to generate more such cells due to stronger phagocytosis. The hybrid cells are more tumorigenic and metastatic, and the hybrid tumors in which immune cell infiltration was inhibited expressed a higher level of Myc. Hybrid cells gained more mesenchymal features, which may enhance their survival in circulation and promote their metastasis. These are very interesting discoveries that would advance the understanding of the composition and function of TME, and targeting such hybrid cells may potentially lead to improved anticancer efficacy.

Despite the enthusiasm, several concerns need to be addressed by the authors.

Major concerns:

1. The clinical implications of the hybrid cells remain unclear. Is there any correlation between the existence or frequencies of the hybrid cells and the prognosis of the patients? What is the impact of the hybrid cells on the efficacy of immunotherapy, e.g., therapy targeting CD47?
2. U937 macrophage model was used in most of the experiments identifying and characterizing the hybrid cells, as well as the in vivo experiments examining tumorigenicity and metastasis. While U937 is a valid model for studying macrophage function, it is a lymphoma cell line. This raised the concern of whether the “hybrid cells” actually derived from macrophage phagocytosis of cancer cells or whether it is an artificial observation of fusions of two types of malignant cells. Primary macrophages should be used to examine and confirm these findings.

3. In Fig2, 3, and Sup Fig2, the authors showed that much less infiltration of tumor-associated macrophages (both M1 and M2) and MDSCs were observed in hybrid tumors. These cells are usually considered pro-tumorigenic components of the TME. This is inconsistent with the findings that hybrid cells are more aggressive and tumorigenic.

4. In Fig5 and Fig7, the presence of the hybrid cells should be further confirmed by cytogenetic analysis, as the experiment showed in Sup Fig2d.

5. In Fig5, 6, and 7, the authors examined the naturally formed hybrid cells in mouse models and patient specimens. It remains unclear, however, whether these hybrid cells are initiated by phagocytosis. To address this, the experiment in Sup Fig4 should be performed with cancer cell models (C4-2, e.g.) with CD47 knockdown and CD47 overexpression to determine whether the susceptibility of cancer cells to phagocytosis correlates with the frequency of hybrid cells.

6. Can the author generate hybrid cells with mouse cells and examine their tumorigenicity and impact on TME in an immunocompetent mouse model?

Minor questions:

1. What are the target cancer cells used in the experiment of Fig1d?

2. Would co-culture of cancer cells and macrophages which express Sirpa show the same effects of treating cancer cells with soluble Sirpa shown in Fig1i-k?

3. Experimental details should be provided about the enrichment of C4-2 hybrid cells and transplantation of the mice in Fig2.

4. In Fig2, the authors showed that hybrid cells might gain hyperproliferation by upregulation of MYC. Is this an intrinsic feature of hybrid cells or extrinsic effects induced by other components in the TME? Comparing hybrid cells (before transplantation) and the hybrid cells isolated from tumors may address this question.

Reviewer #2 (Remarks to the Author): with expertise in macrophage-cancer hybrids

The cancer cell fusion theory as an explanation for metastasis was described by Otto Aichel in the 1900s. However, due to limitations in technology at that time, it was not until decades later that this hypothesis was once again evaluated. Since then, several authors have demonstrated that hybrids between tumor cells and monocytes/macrophages produced in vitro and in vivo had enhanced metastatic potential.

Chih-Wei Chou et al. present an excellent manuscript that thoroughly investigates the fusion theory. This reviewer considers the document to be of great scientific value and answers several of the outstanding questions.

The authors, in agreement with several other researchers in the scientific community, demonstrate the occurrence of fusion between tumor cells and macrophages using numerous in vitro and in vivo models. One of the strengths of this work is the postulation of a mechanism by which fusion is regulated implicating a known signal involved in phagocytosis (CD47). Moreover, the authors describe the potential participation of c-Myc both in providing proliferative capacity to hybrid cells and in the formation of hybrids itself. As an interesting data, Chih-Wei Chou et al. provide evidence indicating the acquisition of "flexibility" by the hybrids, facilitating their mobility. Finally, they attempt to validate part of their results in a limited cohort of patients.

In relation to the postulated mechanism, although Chih-Wei Chou et al. show an interesting series of experiments that support their theory of incomplete phagocytosis as a basis for the generation of hybrids, strong tests of gene overexpression and suppression are missing. The latter also occurs when the authors postulate the potential role that c-Myc plays in this process. Perhaps the greatest weakness of the work is its translation to the reality of patients. The authors have not been exhaustive in the selection and description of the cohort of recruited patients, which is very limited. Furthermore, the measurements carried out on the samples have been scarce in comparison with the exquisiteness of the tests carried out in vivo and in vitro.

All of this leads this reviewer to request that the manuscript be thoroughly revised based on the points listed below before publication.

Majors:

- 1-Due to the importance of CD47 in the postulated mechanism of fusion, a number of experiments in which CD47 expression is modulated are mandatory.
- 2- In line with the latter the same is required for c-Myc.
- 3- There are several hallmarks that have been established for those cells that are potentially metastatic, among them the ability to migrate and immune evasion. The authors postulate that hybrids formed from incomplete phagocytosis may be responsible for metastasis. In this sense, it is necessary to demonstrate both hallmarks: migration capacity and immune evasion. Although the ability to migrate was demonstrated in the in vivo experiments, an illustrative in vitro experiment is missing. In addition, immunological evasion assays are not shown. What happens when hybrids face expanded NKs? Do heterologous CD4 and CD8 lymphocytes proliferate when faced with hybrids? These assays using "parental" cells as controls are necessary to reliably demonstrate immune evasion.
- 4- Other authors have shown that hybrids are generated not between tumor cells and monocytes/macrophages, but between the latter population and tumor stem cells (Cancers (Basel). 2022 Jul 15;14(14):3445. doi: 10.3390/cancers14143445 and Oncoimmunology. 2020 Jun 16;9(1):1773204. doi: 10.1080/2162402X.2020.1773204). Can the authors ensure that the fusions they observe only depend on the expression of CD47 and not on the stemness state of the tumor cells?
- 5- Increasing the size of the patient cohort is a necessity in order to validate the authors' conclusions.
- 6- The discussion should be more discussion and less repetition of the results. It would be recommendable to have a couple of paragraphs where the authors discuss the progress that the data they provided in this manuscript means in the context of fusion theory and the works that have recently been published in this field.

Minors:

- 1- Introduction: Authors said: "These hybrid cells exhibited some characteristics of parental macrophages and cancer cells, similar to those cases previously reported in colorectal, pancreatic, lung, and renal cancers and melanoma". They must cite seminal works where hybrid cells have been identified and described in lung and colorectal cancer: Cancers

(Basel). 2022 Jul 15;14(14):3445. doi: 10.3390/cancers14143445 and Oncoimmunology.

2020 Jun 16;9(1):1773204. doi: 10.1080/2162402X.2020.1773204 among others.

2- Authors said (line 93) "M-CSF-stimulated primary monocytes (Supplementary Fig. 1a)"; however, in this figure they show U937-derived macrophages co-cultured with cancer cells. In the same supplementary figure, they should indicate if it was a confocal microscopy image.

3- A similar experiment conducted with human primary monocytes/macrophages is missing.

4- Due to the fact that some authors have described comparable levels of hybridization between tumor cells and macrophages/monocytes, do authors think that a similar effect could be observed with monocytes instead of macrophages? New data is necessary to answer this question.

5- (Figure 1 e and f) How did authors calculate the percentages of hybrid cells?

6- Some very important questions remain to be answered, among them: Is the population that they define as hybrid polynucleate? Is there fusion between the nuclei of macrophages and tumor cells?

7- What is the reason for the discrepancy between the formation of hybrids between U937 and macrophages from a primary culture?

8- In line 143, authors said: "Most hybrid cells formed in co-culture were difficult to propagate in long-term cell culture"; however, there are reports where isolated tumor hybrid cells showed their proliferative capacity. Authors must cite and discuss these reports.

9- In Supplementary Fig. 2a: The authors should clarify the experiment they did. As I understood it, the hybrids injected into the animals were not isolated and thus contamination with parental cells, mainly tumors, was not avoided. On the other hand, authors ensure that the cells injected into the animals in the "hybrid" experimental condition contained around 50% hybrids. Based on the efficiency shown in the previous figures, that is difficult to obtain; therefore the authors should clarify the methodology followed in this regard. Finally, the authors should explain and discuss why only three of the 10 animals showed tumor growth.

10- In Supplementary Fig. 2: the y-axes of panels a, b, and c should be homogeneous for easy comparison. How do the authors explain the observed growth of the parental condition macrophages?

11- Supplementary Fig. 2: This experiment should be redesigned: after the co-culture

protocol, the resulting populations (hybrids, tumors, and macrophages) should be isolated by sorting and then inoculated into the animals. To do this protocol using primary human monocytes/macrophages isolated from circulation is highly recommended.

12- Chih-Wei Chou et al. show a significant increase in the expression of c-Myc in hybrids, postulating the involvement of this factor in the proliferation of these cells; however, in the presence of an anti-c-Myc antibody, what they observe is a reduction in the generation of hybrids. This observation is very interesting and raises several questions that the authors should address. Crucially, the authors should design some experiment to determine the role of c-Myc in the fusion process. Furthermore, they should clarify the experimental design they followed in this trial: When was the c-Myc inhibitor added? What controls were used in the experiment?

13- In figure Supplementary 5: percentages lacked.

14- Due to the importance of the study and the resounding results obtained in the in vivo and in vitro models, it is mandatory to translate the results to humans using a significant number of patients to validate the authors' conclusions. Unfortunately, the number of patients included in the study is significantly low (only 16). Furthermore, the information provided by the authors on the patients included in the study is scarce, Supplementary Table 5 (age and many other characteristics to be considered are not provided). The latter is essential for the manuscript to be published.

15- The authors identify the hybrids in circulation of patients using the CD45+EPCAM+ signature; however, from their results and the data provided by other authors, the hybrids originate from the fusion between tumor cells and monocytes/macrophages, which indicates that the signature should be refined including some marker from this population, for example: CD14.

Reviewer #3 (Remarks to the Author): with expertise in prostate cancer, macrophages

The authors present an extensive study into the development and characteristics of macrophage-prostate cancer hybrid cells. CD47 expression on cancer cells, is known as a 'don't eat me signal' for macrophages. The authors show that activation of CD47 and downstream signaling results in incomplete digestion of prostate cancer cells. Prostate cancer cells with high expression of CD47 are less phagocytized than cells with a lower CD47

expression. Phagocytosis of prostate cancer cells with low CD47 expression, is incomplete and results in a hybrid cell type, sharing features of macrophages and epithelial cells. These hybrid cells, express both M1 and M2 macrophage differentiation genes, which is regulated by c-MYC. Consequently, the hybrid cells might play a role in immunosurveillance evasion. Moreover, these hybrid cells display anti apoptotic properties and have a role in epithelial-mesenchymal transition and the development of distant metastases. Finally, a relation was assumed in a small patient cohort, between a higher circulating hybrid/CTC ratio and metastatic and castration resistant disease. This relation was essentially described earlier in a small cohort of breast cancer patients (Gast CE et al, Science Adv, 2018). Apart from identification of CD47 signaling as a target for future therapies, levels of circulating macrophage-prostate cancer hybrids might be assessed in liquid-biopsies and serve as a biomarker of disease development.

Although interesting, and extensive studies are presented, the manuscript lacks some focus and with that, new mechanistic insights in the regulation of properties of the macrophage-prostate cancer cell hybrids. Moreover, after decades of studies into hybrids of macrophages and cancer cells, one would expect that the clinical relevance of these hybrids has been studied in more detail. The present study contributes only limited additional evidence for relevance of studies into this subject.

Major comments

1. The proposed model of the development of macrophage-prostate cancer hybrid cells is predominantly based on a highly artificial in vitro model. Although three prostate cancer cells are included in the studies, most in vitro work is done in the C2-4 prostate cancer cells with the lower CD47 expression and development of hybrids. Moreover, predominantly one model of human macrophages is used, the human U937 derived macrophage like cells. M-CSF stimulated monocytes and murine IC-21 derived macrophages are only used for a limited number of studies. Hybrid cells of primary macrophages with C2-4 cells are rarely found, while none are formed with DU-145 and 22RV1 cells (Fig 1e). It would be very helpful if critical findings could be confirmed in an additional model of prostate cancer and macrophages.

2. The concept that fusion cells with an abnormal chromosome distribution contribute to cancer development is a long-standing theory, of which supporting data was summarized in a 2008 Nat Rev Cancer publication. However, this field of research remains an in vitro exercise for decades, apart from a few small cohorts of cancer patients where occurrence of circulating fusion cells was associated with survival (including the present manuscript). In the current manuscript the presence of hybrids in 11 primary prostate cancer was assessed by scRNA sequencing (multiple macrophage and epithelial markers). This is strong, but the question remains what the relevance is of the macrophage-prostate cancer hybrid cells in human prostate cancer? Do they occur in primary prostate tumors and metastases? How big is the interpatient and inpatient variation? Is their presence associated with outcome? Although supporting the in vitro findings, the presented translational studies are weak, due to low number of patients and definition of compared cohorts. A relation between multiplex immunohistochemistry staining for markers of hybrid cells in a large cohort of primary and metastatic prostate cancer lesions (TMA), in relation to clinical outcomes (time to metastases, biochemical recurrence, progression free survival) would significantly enhance the relevance of the studies.

Minor comments

1. High CD47 expression is associated with advanced cancers and suppressed phagocytosis by myeloid cells. How do the authors explain the association between low CD47 expression and a shorter time to biochemical recurrence in the TCGA dataset (Figure 1a)? Moreover, there seems to be no difference in expression of CD47 between benign prostate tissue and primary prostate cancer, while expression is lower in metastatic prostate cancer. This finding seems to be very consistent in three individual datasets (Figure 1b). How is this explained?

2. The claim that macrophages and prostate cancer cells fuse to a hybrid, is based on expression of a single macrophage and an epithelial marker. This claim can be made a lot stronger. Some cytogenetic analysis is presented in Suppl Fig 2d, but how many chromosomes do the macrophage like cells, the prostate cancer cells and the hybrids have?

3. It should be clear in all figures/legends, which model of macrophages has been used.
4. It is surprising that the biggest effect of blocking CD47 (B6H12) on hybrid formation is in the cells with the lowest CD47 expression (C4-2). Suppl Fig 1G. how is this explained?
5. Number of genes assessed to establish M1 and M2 polarization of the macrophages is really limited. No single gene expression confirms differentiation unequivocally. Montovani et al described multiple markers of M1 and M2 polarization. The claim that the hybrid cells adopt both polarizations should be substantiated with a wider panel of markers.
6. AR signaling is the single most important growth signal in prostate cancer. All three prostate cancer cell lines assessed express AR and are responsive to testosterone. Moreover, also macrophages express AR. Do the hybrid cells express AR? Studies into the functionality of AR signaling in the hybrid cells would be really interesting and relevant.
7. Also, the rare putative macrophage-prostate cancer cells hybrids in human circulation (0,21% of PBMCs) are identified by one immune cell (and not macrophage) marker (CD45) and one epithelial marker (EpCAM). Considering the limited number and nonspecific markers and its very low frequency, these findings might concern other hybrids of immune and epithelial cells, but also be an artifact.

Reviewer #4 (Remarks to the Author): with expertise in prostate cancer, macrophages

In the study titled “phagocytosis, initiated hybrid tumor cells acquire c-myc mediated quasi polarization state for immunoevasion and distant dissemination”, authors have studied the role of macrophage-induced phagocytosis as a defense mechanism and how the cancer cells, which expresses the cd47 repel the macrophage engulfment. Moreover, the authors have suggested that the low-grade prostate cancer cells express low cd47 and, as a result of which, they are more prone to phagocytosis. On the contrary, the high-grade prostate cancer cells, such as 2RV1 and DU145 express high cd47, which promotes the formation of hybrid cells which imitate the dual macrophage features and require c-myc for their phenotypic properties and present as a new target for the treatment of the disease,

There are several limitations in this study; these are as follows-

In lines 120 to 122. The author suggested that “do not eat me signals upon treatment with CD24- Siglec 10, and PDL1-PD1 does not exert an apparent effect compared to CD47-SIRPalpha in modulating the hybrid cell formation”. The question is, what specific molecular changes are induced by CD47-SIRPalpha that make eat me, not process, specifically dominant in high-grade prostate cancer cells?

In lines 127 to 129, the author suggested that an “increased amount of bcl2 protein may enhance the apoptotic activities to prevent the complete digestion of entrapped cancer cells inside the macrophages”, Did the authors compare the extent of BCL increase in other PCa cell types post-exposure with CD24- Siglec 10, and PDL1-PD1?

Additionally, Oxidative stress increase or the ROS level increase can also influence the severity of the disease. I wonder if the authors evaluated if the CD47-SIRPalpha influences oxidative stress, which could potentially explain the extent of differences between the effects of CD24- Siglec 10, and PDL1-PD1 and CD47-SIRPalpha.

In lines 144 to 147, the authors suggested that “the hybrid cells evasion can influence the immune surveillance and gaining hyperproliferative growth in the host.” For this, the authors used nude mice. Though they admit that the nude mice lack the essential components of the immune system like T and B cells, they used these murine models to focus on macrophage polarization. The concern arises when authors use the outcomes from these models to generalize them to make claims on the influence of their findings on the overall immune surveillance, which is not true. It is strongly recommended to either remove these claims or use C57Bl6 mice grafted with TRAMP2 cells as models to generate hybrid cells.

In line, 158 to 160 author suggested that “to determine the transcriptomic composition of hybrid cells and their impact on host immune microenvironment they perform special expression profiling on hybrid and Parental tumor sections.” Once again, the murine models used in this study are limited because they are insufficient to depict anything in the tumor

microenvironment. Secondly, in the transcriptomic analysis, the author suggested that “the observed human and murine genes such as testing the macrophage infiltration.” In these tumor areas, did the authors compare the difference in the infiltration rate or the type of M1/M2 macrophages from the murine models in the context of the cell types (non-hybrid versus hybrid) that were grafted?

Inline 195 to 197, the authors suggested that “hybrid cells are likely benefited from the quasi macrophage characteristic to evade further the surveillance of other immune cells in the host”. The question is, did the authors explore any specific aspects experimentally, highlighting that the quasi-macrophage characteristics enhance the evasion of MDSCs and natural killer cells?

In the segment, “hybrid cells acquire C-MYC modulated m1 to m2 gene transcription to achieve distinct dissemination,” did authors evaluated the efficacy of C-MYC inhibition in reducing the functional capabilities (reducing the bcl2 expression of hybrid cells), specifically in the context of other, do not eat me modulators such as CD24- Siglec 10, and PDL1-PD1 and CD47-SIRPalpha.

Also, did the authors compare the changes in the peaks in the c-Myc binding regions in low-grade versus high-grade prostate, cancer cells, or the patients?

Inline 236 to 239 author suggested that “to determine the impact of hybrid cells on metastasis, they isolated the cells from spine and femur fragments of the xenograft”. The questions are (1) Did the authors use the spine and femur from hybrid cells, C42B, 22RV1, and DU145 cells? (2) Did the authors evaluate the changes (if any) in the c-Myc expression in these?

In the segment “Low CD47 expressing primary tumors are more prone to generate hybrid cells”, did the authors evaluated if the artificially generated hybrid cells could act as a modulator to induce the prevalence of naturally occurring hybrid cells? Secondly. Did the authors compare the extent of hybrid cells that are naturally occurring by grafting them in mice with non-hybrid or hybrid cells from C42B, 22RV1, and DU145 cells?

The authors mentioned that a customized Seurat script was used, and 30 principle components were used in the analysis. Is the code readily available for analysis in an RMarkdown file or R script? Also, did the authors optimize the analysis for 30 PC selection and resolution tuning?

Hallmark gene sets were used, and then customized gene sets from the literature. Could the authors have used some of the other immune-specific gene sets within MSigDB?

The authors mentioned using GSE176031 from the GEO repository and filtering out cells of low quality. How many cells were filtered out with their criteria, and how many were kept?

Line 761 authors mentioned using FindAllMarkers in Seurat and the corresponding p-value is given by Bonferroni correction, but this should be the `p_val_adj`. Also, was there any consideration given to FC when looking at the differentially expressed genes?

Reviewer #5 (Remarks to the Author): with expertise in single-cell multi-omics and spatial technologies

In this manuscript, Chou et al. experimentally and computationally studied the formation of tumor-macrophage hybrid cells and investigated the characteristics of these hybrid cells. They first conducted phagocytosis assays in prostate cancer cell lines and found that low CD47-SIRP α signal is associated with hybrid cell formation. Then they observed that hybrid cells acquire c-Myc-modulated M1/M2 gene transcription to evade immunosurveillance. They also found that these hybrid cells could enhance mesenchymal features to achieve distant dissemination. Overall, this study is comprehensive and addresses an important issue that might be of interest to a broad international audience. However, there are several important points which need to be addressed and clarified.

(1) The results in this study were mainly generated based on prostate tumor, but the authors didn't mention this point in the abstract and introduction. Do the conclusions inferred from prostate tumor also apply to other types of cancer?

(2) In Figure 2b, how did the authors define the minor regions (1a, 3a and 8a)? It would be

better to label these minor regions to t-SNE map. The number of spots in these regions are small, especially for 1a, it would be better to discuss the robustness of defining these minor regions.

(3) Line 174-176 says 'hybrid tumor usually expressed higher human genes (e.g., VIM, TUBB, and MYC) and lower murine genes (e.g., Vim, Col1a1, and Ctsb) compared to the parental tumor', but Figure 2d/e just showed the expression in a subsection but not the whole region, whether is there significant difference between these two tumors on the overall level?

(4) In Figure 3e, the authors should do the statistical test between DMSO and F4 in level of replicate samples, like Figure 3d. Different replicate samples are likely to be considered as the independent variables. Figure 4i/j, Figure 6 and Figure 7 regarding comparison of gene expression between different groups would also be updated.

(5) Figure 3H indicated the role of c-Myc in enhancing the transcription of genes related to macrophage polarization by ChIP. It would be better to valid these results by qPCR or Western Blotting as a supplementary file.

(6) In Line 279, the authors stated that putative hybrid cells were identified based on the co-expression of epithelial and macrophage marker genes, so how to choose the threshold to determine a cell which expresses epithelial genes or macrophage genes, the detailed procedure of selecting putative hybrid cells from scRNA-seq data should be added in the Methods. The putative hybrid cells are formed by two cells, did these cells have relative higher gene numbers or umi counts?

(7) Following the above point, how do the authors prove that the defined hybrid cells from scRNA-seq is the real hybrid cells but not the doublets, as the doublets is very common in scRNA-seq data? Did the authors try methods of rare cell type identification, do those real hybrid cells show specific gene programs?

(8) In Figure 1D, the authors used CD86 (M1 markers) as the marker of putative hybrid cells. However, in Figure 7A, they used CD163 (M2 markers) to identify hybrid cells. What is the reason for this inconsistency?

(9) There are some redundant plots in figures, it would be good to move them to supplementary figure. For example, figure 1e and 1f.

(10) Lines 176-178, the authors stated that "Four of the seven main regions (i.e., #1, 5, 6, and 8) in the parental tumor contained high levels of murine genes while all hybrid tumor

regions expressed elevated human genes (Fig. 2f)", it looks that the % of human genes in region 5 of parental tumor is higher, and also the region 1, is it possible to provide the p values for the comparison?

Point-by-point response

Reviewer #1

Major comments:

1. The clinical implications of the hybrid cells remain unclear. Is there any correlation between the existence or frequencies of the hybrid cells and the prognosis of the patients? What is the impact of the hybrid cells on the efficacy of immunotherapy, e.g., therapy targeting CD47?

Response: Since the first submission, we have extended two clinical datasets to this study. First, IHC study of 50 primary and 44 metastatic tissue sections from 89 prostate cancer patients revealed that hybrid cells were preferentially found in metastatic tumors relative to primary tumors and patients with higher Gleason scores or older ages (**Fig. 5j-o**). Second, gene profiling of four metastatic castration-resistant prostate cancer (mCRPC) cohorts (total n = 634) suggests that hybrid cells are linked to an adenocarcinoma-immune subtype of mCRPC. Accordingly, we have revised the manuscript by including these new findings in the main text (**Line 293-300 and 316-333**).

Our finding that low CD47 signals render incomplete phagocytosis and hybrid cell formation has an important ramification, particularly in targeting this “don’t-eat-me” signal for immunotherapy. While the anti-CD47 treatment can promote the phagocytic uptake of cancer cells by macrophages, the unintended consequence is that partial phagocytosis may occur, giving rise to tumor-macrophage hybrid cells. In **Line 425-427**, we have stated that *“.....Therefore, our present study highlights a potentially overlooked effect of immunotherapy, presenting opportunities to investigate new agents targeting the Achilles’ heel of hybrid cells.”*

2. U937 macrophage model was used in most of the experiments identifying and characterizing the hybrid cells, as well as the in vivo experiments examining tumorigenicity and metastasis. While U937 is a valid model for studying macrophage function, it is a lymphoma cell line. This raised the concern of whether the “hybrid cells” actually derived from macrophage phagocytosis of cancer cells or whether it is an artificial observation of fusions of two types of malignant cells. Primary macrophages should be used to examine and confirm these findings.

Response: In addition to using U937-derived macrophages, we have now included primary human and murine macrophages in our co-cultured studies (see **Fig. 1e-right, Supplementary Fig. 5**). The overall efficiency of forming hybrid cells was lower when primary macrophages were used in co-culture compared to U937 macrophages (**Fig. 1e-right**). We have attributed this low efficiency in part to the high digestion efficiency of cancer cells by primary macrophages (see the proportion of C4-2 cancer cells was reduced from 40% at 2 hr to ~2% at 72-hr in co-culture, **Supplementary Fig. 1e-left; Line 107-111**). However, the overall frequency of hybrid cell formation remained the same between IC-21 and primary macrophages in murine models (**Supplementary Fig. 5**). Therefore, more studies need to be done to determine how hybrid cells are formed in a cellular context-dependent manner.

Although we could not rule out the possibility of spontaneous fusion between a cancer cell and a macrophage in our co-culture model, we include two lines of experimental evidence that reasonably support that hybrid cells can be derived from an incomplete phagocytosis event. In the revised manuscript, we have stated that *“.....Recently, two populations of hybrid cells were characterized - circulating hybrid cells (CHCs) and cancer-associated macrophage-like cells (CAMLs). The former can result from spontaneous fusion, while the latter is considered a “macrophage masquerader” containing incompletely digested tumor materials. Our present study suggests that most hybrid cells generated in co-culture fit the description of CAMLs. First supporting evidence comes from our karyotypic analysis of metaphase spreads that hybrid cells had hyperdiploid chromosome numbers slightly larger than that of parental macrophages (see **Supplementary Fig. 4d**). If hybrid cells were derived from spontaneous fusions, we assume that their nuclei have polyploid chromosomes acquired from both parental cells. Second, our CD47-SIRPα study implies that weak “don’t-eat-me” signals can facilitate the formation of CAMLs through incomplete phagocytosis.”*(**Line 409-418**).

3. In Fig2, 3, and Sup Fig2, the authors showed that much less infiltration of tumor-associated macrophages (both M1 and M2) and MDSCs were observed in hybrid tumors. These cells are usually considered pro-tumorigenic components of the TME. This is inconsistent with the findings that hybrid cells are more aggressive and tumorigenic.

Response: There are two major immune arms that dictate the balance of the TME, favoring or suppressing tumor cell growth. While M2-like macrophages and MDSCs belong to the pro-tumorigenic arm, M1-like macrophages and other types of immune cells (e.g., NK cells) are anti-tumorigenic in the TME. The reduced infiltration of overall immune cells in the hybrid tumor relative to the parental tumor suggests the former being a “cold” tumor. We believe that hybrid cells acquire both M1- and M2-like macrophage features to confound the host’s immunosurveillance. In the previously submitted manuscript, both c-Myc and M1- M2-macrophage genes were shown to be highly expressed in hybrid tumor. A new set of RT-qPCR data has strengthened the evidence that c-Myc in hybrid cells could regulate this M1- and M2-like characteristics (**Fig. 3g**). Furthermore, shRNA knockdown of *MYC* in hybrid cells could unmask their immunoevasive camouflage characteristics, tipping the immune balance towards a more anti-tumorigenic TME (**Fig. 3h, Line 214-225**). In the revised manuscript, we have stated that “...*These spatial mapping data suggest that hybrid cells can benefit from the quasi-macrophage characteristics as camouflage to evade the surveillance of murine macrophages or other myeloid cells, including pro-tumorigenic myeloid-derived suppressor cells (MDSCs) and anti-tumorigenic natural killer (NK) cells ...*” (**Line 182-185**).

4. In Fig5 and Fig7, the presence of the hybrid cells should be further confirmed by cytogenetic analysis, as the experiment showed in Sup Fig2d.

Response: We appreciate this suggestion in clinical samples. Due to the scarcity (0.2%; **Fig. 8c**) of hybrid cells in patients’ blood samples, this poses a tremendous challenge to obtain and expand them for routine clinical cytogenetic analysis. The recent development of microfluidic devices (Descamps et al., *Int J Mol Sci* 2022) offers a new opportunity to isolate and expand these rare cells from blood samples. As such, we may adapt this *ex vivo* procedure for our cytogenetic studies in the future. However, here we have provided two assays to study hybrid cells in patients’ blood samples - 1) microfiltration coupled with immunofluorescence imaging and 2) single-cell proteomic CyTOF (**Fig. 8**).

5. In Fig5, 6, and 7, the authors examined the naturally formed hybrid cells in mouse models and patient specimens. It remains unclear, however, whether these hybrid cells are initiated by phagocytosis. To address this, the experiment in Sup Fig4 should be performed with cancer cell models (C4-2, e.g.) with CD47 knockdown and CD47 overexpression to determine whether the susceptibility of cancer cells to phagocytosis correlates with the frequency of hybrid cells.

Response: We have followed the reviewer’s suggestion and performed *CD47* knock-in and -down experiments and related co-culture assays. The new data further support that weak anti-phagocytic *CD47* renders incomplete phagocytosis and hybrid cell formation (**Fig. 1f, g, Line 110-111**).

6. Can the author generate hybrid cells with mouse cells and examine their tumorigenicity and impact on TME in an immunocompetent mouse model?

Response: In response to the important issue, we have generated hybrid cells using two syngeneic prostate cancer cell lines, high *Cd47*-expressing RM1 and low *Cd47*-expressing TRAMP-C2, which were further co-cultured with murine IC-21 or primary macrophages (**Supplementary Fig. 5, Line 277-279**). To test their tumorigenicity, we injected RM1 or TRAMP-C2 cells through the tail vein of immunocompetent C57BL/6J male mice. The new finding has now been described in the text “...*we compared their rates of hybrid cell formation in vivo. Although invading RM-1 cells formed large lesions in the lung, the incidence of hybrid cell formation was low (~2%), reflecting the in vitro finding (Fig. 5d-f). In contrast, invading TRMAP-C2 cells developed micrometastasis in the lung but displayed hybrid cells in ~30% of the lesions (Fig. 5g-i).*” (**Line 279-283**).

Minor comments:

7. What are the target cancer cells used in the experiment of Fig1d?

Response: We have revised flow cytometry diagrams shown in the original Fig. 1d. The panels and relabeled cell line names have now been provided in new **Fig. 1d, Supplementary Fig. 1d, and Supplementary Fig. 5b, e**.

8. Would co-culture of cancer cells and macrophages which express Sirpa show the same effects of treating cancer cells with soluble Sirpa shown in Fig1i-k?

Response: We have since performed a new experiment by directly adding SIRP α to the co-culture. The result showed "...Despite BCL-2 levels being enhanced in parental cancer cells and macrophages, the induction was again more pronounced in hybrid cells (**Supplementary Fig. 1l, m**)..." (Line 122-123).

9. Experimental details should be provided about the enrichment of C4-2 hybrid cells and transplantation of the mice in Fig2.

Response: The illustrations elucidating hybrid cell enrichment, cell inoculation, and spatial gene profiling have been updated in revised **Fig. 2a**.

10. In Fig2, the authors showed that hybrid cells might gain hyperproliferation by upregulation of MYC. Is this an intrinsic feature of hybrid cells or extrinsic effects induced by other components in the TME? Comparing hybrid cells (before transplantation) and the hybrid cells isolated from tumors may address this question.

Response: We thank the reviewer for this suggestion to tease out intrinsic and extrinsic factors contributing to the growth of hybrid cells. We suggest that overexpressed c-Myc is an intrinsic factor, endowing hybrid cells with immunoevasive and proliferative potential in the host TME. While the present focus is on c-Myc, future RNA-seq can be performed on hybrid cells before and after exposure to the host TME *in vivo* to identify extrinsic factors that activate downstream oncogenic and immune pathways in these cells.

Reviewer #2

Major comments:

11. Due to the importance of CD47 in the postulated mechanism of fusion, a number of experiments in which CD47 expression is modulated are mandatory.

Response: The CD47 knock-in and -down experiments have been performed per the reviewer's request. The new results are presented in **Fig. 1f, g**. Accordingly, the main text has been revised "...Co-culture experiments using CD47 knock-in or -down cells partially decreased or increased hybrid cell formation, respectively ..." (**Line 110-111**).

12. In line with the latter the same is required for c-Myc.

Response: MYC shRNA knockdown (KD) has been performed using CWC cell lines derived from hybrid tumors (**Fig. 3f**). Hybrid characteristics of CWC cells have been authenticated by the co-expression of dual macrophage-epithelial markers and the co-presence of parental marker chromosomes. The new finding of this KD study has now been described in the text "...This shRNA KD resulted in the downregulation of M1- and M2-like genes, thus suggesting the immunoregulatory role of c-Myc in hybrid cells (**Fig. 3g**)..." (**Line 211-213**).

13. There are several hallmarks that have been established for those cells that are potentially metastatic, among them the ability to migrate and immune evasion. The authors postulate that hybrids formed from incomplete phagocytosis may be responsible for metastasis. In this sense, it is necessary to demonstrate both hallmarks: migration capacity and immune evasion. Although the ability to migrate was demonstrated in the in vivo experiments, an illustrative in vitro experiment is missing. In addition, immunological evasion assays are not shown. What happens when hybrids face expanded NKs? Do heterologous CD4 and CD8 lymphocytes proliferate when faced with hybrids? These assays using "parental" cells as controls are necessary to reliably demonstrate immune evasion.

Response: Our spatial gene profiling suggests that "...hybrid cells can benefit from the quasi-macrophage characteristics as camouflage to evade the surveillance of murine macrophages or other myeloid cells, including pro-tumorigenic myeloid-derived suppressor cells (MDSCs) and anti-tumorigenic natural killer (NK) cells (**Supplementary Fig. 3i-j**)" (**Line 182-185**). Further studies also implicate that c-Myc plays a role in the immunomodulatory function of hybrid cells:

Following the reviewer's suggestion, we have performed immunological evasion assays by co-culturing "cultured CWC cells carrying control or MYC shRNA KD with human peripheral blood mononuclear cells (PBMCs) subsequently stimulated with mitogenic lectins to activate T cells. Flow cytometry analysis was performed to identify cytotoxic CD8⁺ T cells, regulatory CD25⁺CD4⁺ T cells, and naïve CD25⁻CD4⁺ T cells after the co-culture (**Supplementary Fig. 4f**). The proportion of cytotoxic T cells showed a 2-fold increase in PBMCs co-cultured with control CWC cells relative to PBMCs only (**Fig. 3h-left**). This T cell population increased from 1.8 to 3.8% after co-culturing with CWC cells carrying MYC shRNA KD. An increase (60%) of regulatory T cells occurred in PBMCs co-cultured with control CWC cells relative to PMBCs only (**Fig. 3h-middle**). Nevertheless, this induction was reduced by exposing PBMCs to cells carrying MYC shRNA KD. The shRNA KD had no effect on naïve CD25⁺CD4⁺ T cells (**Fig. 3h-right**). Collectively, this experiment suggests that MYC knockdown may unmask immunoevasive characteristics of CWC cells, fostering more involvement of cytotoxic T cells to initiate an anti-tumorigenic microenvironment" (**Line 214-225**).

Since we focus on the regulatory role of c-Myc, CWC cells carrying scrambled shRNA were used as control.

14. Other authors have shown that hybrids are generated not between tumor cells and monocytes/macrophages, but between the latter population and tumor stem cells (Cancers (Basel). 2022 Jul 15;14(14):3445. doi: 10.3390/cancers14143445 and Oncoimmunology. 2020 Jun 16;9(1):1773204. doi: 10.1080/ 2162402X. 2020. 1773204). Can the authors ensure that the fusions they observe only depend on the expression of CD47 and not on the stemness state of the tumor cells?

Response: We have since performed two sets of experiments by co-culturing 1) cancer cells with U397 or primary monocytes and 2) stem-like side population of cancer cells with macrophages. These results are presented in **Supplementary Fig. 1e, f** and the revised main text "Hybrid cell formation appeared more efficient in cancer cells co-cultured with differentiated macrophages than those with parental monocytes..." (**Line 100-101**).

In line with the previous findings by Montalbán-Hernández et al., *Cancers* 2022 and Aguirre et al., *Oncoimmunol* 2020, we found that “the main source of hybrid cells was likely derived from stem-like side populations of a cancer cell line...” (Line 101-103). We speculate that these stem-like side populations express low CD47 and, therefore, are more susceptible to incomplete phagocytosis and hybrid cell formation. We sincerely thank the reviewer for this idea that provides a future research direction.

15. Increasing the size of the patient cohort is a necessity in order to validate the authors' conclusions.

Response: The comment is well taken, and we have since performed an immunohistochemical analysis of a Taiwanese cohort of 50 primary prostate tumors and 44 metastatic tumors (Fig. 5j-p). Extending our initial analysis of 15 tumors and normal controls (Supplementary Fig. 6a-h), we have conducted gene-signature profiling of four metastatic castration-resistant prostate cancer (mCRPC) cohorts (total n = 634). This finding suggests hybrid cells resembling an mCRPC subtype (Fig. 6, Line 316-333).

16. The discussion should be more discussion and less repetition of the results. It would be recommendable to have a couple of paragraphs where the authors discuss the progress that the data they provided in this manuscript means in the context of fusion theory and the works that have recently been published in this field.

Response: The first paragraph of Discussion has been extensively revised by describing the theory of cancer cell fusion and two cell types - circulating hybrid cells (CHCs) and cancer-associated macrophage-like cells (CAMLs). In addition, we have provided explanations that the majority of our hybrid cells generated in co-culture fit the description of CAMLs (see Line 409-418).

Minor comments:

17. Introduction: Authors said: “These hybrid cells exhibited some characteristics of parental macrophages and cancer cells, similar to those cases previously reported in colorectal, pancreatic, lung, and renal cancers and melanoma”. They must cite seminal works where hybrid cells have been identified and described in lung and colorectal cancer: *Cancers* (Basel). 2022 Jul 15;14(14):3445. doi: 10.3390/cancers14143445 and *Oncoimmunology*. 2020 Jun 16;9(1):1773204. doi: 10.1080/2162402X.2020.1773204 among others.

Response: We apologize for the two omissions, and we have since revised the Introduction section by including these significant contributions of the previous work in Introduction (Line 72-74).

18. Authors said (line 93) “M-CSF-stimulated primary monocytes (Supplementary Fig. 1a)”; however, in this figure they show U937-derived macrophages co-cultured with cancer cells. In the same supplementary figure, they should indicate if it was a confocal microscopy image.

Response: We thank the reviewer for pointing out the confusion. In the original Supplementary Fig. 1a, we used co-cultured U937-derived macrophages and C4-2 prostate cancer cells as an illustration. We have since replaced the illustration using primary macrophages (revised Supplementary Fig. 1d). We have also replaced one confocal microscopy image with time-lapse photos taken by IncuCyte (new Supplementary Fig. 1a). The corresponding figure legend has been rephrased to improve the clarity of description.

19. A similar experiment conducted with human primary monocytes/macrophages is missing.

Response: We have since conducted co-culture experiments using primary monocytes and primary macrophage, as shown in Supplementary Fig. 1e, Fig. 1e-right, and Line 100-101.

20. Due to the fact that some authors have described comparable levels of hybridization between tumor cells and macrophages/monocytes, do authors think that a similar effect could be observed with monocytes instead of macrophages? New data is necessary to answer this question.

Response: A new experiment has since been performed by co-culturing U937 or primary monocytes with C4-2 prostate cancer cells. When comparing the results with those of differentiated macrophages, we found that “Hybrid cell formation was more efficient in cancer cells co-cultured with differentiated macrophages than those with monocytes (Fig. 1e, Supplementary Fig. 1e)” (Line 100-101).

21. (Figure 1 e and f) How did authors calculate the percentages of hybrid cells?

Response: The percentages of hybrid cells were calculated based on the number of hybrid cells per total vital cells (see new **Fig. 1e**, as well as other related figure panels). We have also updated the description in the Method section (**Line 565-568**).

22. Some very important questions remain to be answered, among them: Is the population that they define as hybrid polynucleate? Is there fusion between the nuclei of macrophages and tumor cells?

Response: The majority of hybrid cells generated in co-culture were mononuclear, assimilating two genomes into a synkaryon (see examples in **Supplementary Fig. 1c**). Our karyotypic analysis supports that hybrid cells shared marker chromosomes from both parental macrophages and cancer cells (**Supplementary Fig. 4d, e**). However, we did observe binucleated hybrid cells disseminated in the bone of xenograft mice (**Fig. 4c-upper**). We also found a putative bi-nucleated hybrid cell in a lung metastatic tissue section of a prostate cancer patient (**Fig. 5j-left**). In the Discussion section, we have now described that “...*This heterotypic fusion generates a hybrid cell by assimilating two nuclei into a synkaryon or keeping two or more nuclei as a heterokaryon.....*” (**Line 407-408**).

23. What is the reason for the discrepancy between the formation of hybrids between U937 and macrophages from a primary culture?

Response: Based on our limited co-cultured studies, we observed that primary human macrophages have a stronger ability to perform phagocytosis than U937-derived macrophages. In this case, the high digestion efficiency of primary macrophages greatly decreases the chance of hybrid cell formation in low CD47-expressing C4-2 cancer cells (see **Supplementary Fig. 1g-left, lane #4 and #5, green; Line 107-111**). However, the overall frequency of hybrid cell formation was not overly different when either IC-21 or primary murine macrophages were co-cultured with low Cd47-expressing TRAMP-C2 cells (**Supplementary Fig. 5**). Further research is required to be done to determine how hybrid cells are formed in a cellular context-dependent manner.

24. In line 143, authors said: “Most hybrid cells formed in co-culture were difficult to propagate in long-term cell culture”; however, there are reports where isolated tumor hybrid cells showed their proliferative capacity. Authors must cite and discuss these reports.

Response: Since the first submission, we have successfully derived and maintained hybrid cells in long-term culture (**Supplementary Fig. 4b, c**) and decided to delete the original statement that “*Most hybrid cells formed in co-culture were difficult to propagate in long-term cell culture*”. In the revised manuscript, we have cited two papers (Science Advances. 2018 Sep 12;4(9):eaat7828. doi: 10.1126/sciadv.aat7828. and 2015 Aug 12;10(8):e0134320. doi: 10.1371/journal.pone.0134320) and stated that “...*These hybrid cell lines exhibited proliferative capacity in long-term culture similar to those previously reported...*” (**Line 209-210**).

25. In Supplementary Fig. 2a: The authors should clarify the experiment they did. As I understood it, the hybrids injected into the animals were not isolated and thus contamination with parental cells, mainly tumors, was not avoided. On the other hand, authors ensure that the cells injected into the animals in the “hybrid” experimental condition contained around 50% hybrids. Based on the efficiency shown in the previous figures, that is difficult to obtain; therefore the authors should clarify the methodology followed in this regard. Finally, the authors should explain and discuss why only three of the 10 animals showed tumor growth.

Response: The legend of **Fig. 2a** have been revised to clarify the methodology that “*Nu/Nu mice were subcutaneously inoculated with parental C4-2 cells or a 7-day co-cultured mixture of C4-2 cells and U937 macrophages, which contained ~50% putative hybrid cells in the live cell population...*” (**Line 1172-1174**).

In the revised Discussion section, we have provided an explanation regarding the modest incidence of visible tumor growth in our xenograft study “*Hybrid cells newly formed in co-culture probably contained diverse and unstable fusogenomes, and therefore only a small fraction of viable hybrid cells were selected for propagation as tumor xenografts. As a result, we observed 30% of visible tumor growth in mice transplanted with co-culturally enriched hybrid cells. Although purified hybrid cells were not used for xenotransplantation, parental cancer cells or macrophages remaining in co-culture mixtures could be used as internal controls for growth comparisons. Indeed, our spatial gene profiling revealed that 98.5% of hybrid tumor regions harbored hybrid cells, while the remaining 1.5% contained parental cancer cells (see 1a and 8a in **Supplementary Fig. 2e**). Parental macrophages were not detectable in these regions. The finding suggests that hybrid cells were outcompeting parental cancer cells with a growth advantage in the host microenvironment*” (**Line 428-437**).

26. In Supplementary Fig. 2: the y-axes of panels a, b, and c should be homogeneous for easy comparison. How do the authors explain the observed growth of the parental condition macrophages?

Response: We thank the reviewer's suggestion and have revised the Y-axes of tumor growth curves with identical scales in **Supplementary Fig. 2a-c**. In the main text, we have provided the explanation and cited an article (Clinical Epigenetics, 2018 Nov 8;10(1):139. doi:10.1186/s13148-018-0563-3) that "...*Inoculated parental macrophages had negligible growth in mice, except for one with a visible lump, possibly a lymphoma developed from residual U937 monocytes non-responsive to PMA stimulation (Supplementary Fig. 2c)*" (Line 142-144).

27. Supplementary Fig. 2: This experiment should be redesigned: after the co-culture protocol, the resulting populations (hybrids, tumors, and macrophages) should be isolated by sorting and then inoculated into the animals. To do this protocol using primary human monocytes/macrophages isolated from circulation is highly recommended.

Response: The point raised by the reviewer is well-taken. In the revised manuscript, we have stated that "...*To improve the current condition, we can purify hybrid cells based on flow cytometry sorting for orthotopic inoculation in humanized mouse models, which will be better suited for assessing the growth and metastatic potential of hybrid cells*" (Line 439-441). We also plan to use macrophages derived from primary monocytes to generate hybrid cells in co-culture, albeit with a low efficiency. An alternative is to purify naturally occurring hybrid cells in metastatic sites for long-term cultivation (see Fig. 5g-i). We thank the suggestion for using primary monocytes/macrophages from circulation for our future studies.

28. Chih-Wei Chou et al. show a significant increase in the expression of c-Myc in hybrids, postulating the involvement of this factor in the proliferation of these cells; however, in the presence of an anti-c-Myc antibody, what they observe is a reduction in the generation of hybrids. This observation is very interesting and raises several questions that the authors should address. Crucially, the authors should design some experiment to determine the role of c-Myc in the fusion process. Furthermore, they should clarify the experimental design they followed in this trial: When was the c-Myc inhibitor added? What controls were used in the experiment?

Response: We have since performed the MYC shRNA knockdown study, which significantly supports the role of c-Myc in promoting immunoevasive features of hybrid cells. However, it will require extensive efforts to study the new role of c-Myc in the fusion process, which is beyond the scope of the current study. Therefore, we have decided not to present our preliminary data using an anti-c-Myc antibody to reduce the efficiency of hybrid cell formation in our current manuscript.

29. In figure Supplementary 5: percentages lacked.

Response: We have since revised the gating strategy and provided the percentage of each type in a pie chart (Revised Fig. 7b, c).

30. Due to the importance of the study and the resounding results obtained in the in vivo and in vitro models, it is mandatory to translate the results to humans using a significant number of patients to validate the authors' conclusions. Unfortunately, the number of patients included in the study is significantly low (only 16). Furthermore, the information provided by the authors on the patients included in the study is scarce, Supplementary Table 5 (age and many other characteristics to be considered are not provided). The latter is essential for the manuscript to be published.

Response: Two sets of clinical data with more than 700 prostate cancer patients have been included to study the frequency and characteristics of hybrid cells, please see our response in Point #15. In addition, the age of individual patients has been added to the revised **Supplementary Table 7**.

31. The authors identify the hybrids in circulation of patients using the CD45+EPCAM+ signature; however, from their results and the data provided by other authors, the hybrids originate from the fusion between tumor cells and monocytes/macrophages, which indicates that the signature should be refined including some marker from this population, for example: CD14.

Response: We have replotted our CyTOF to clarify the signature of hybrid cells, which expressed comparable levels of CD14 with those of monocytes/macrophages (**Supplementary Fig. 9a**). In the revised text, we have

stated that “...CyTOF profiling of ~2.9 million PBMCs identified ten main populations based on differential expression patterns of different myeloid and epithelial markers (**Fig. 8b, c, Supplementary Fig. 9a, b, Supplementary Table 7**). Two minor populations - CTCs ($CD45^-/CD3^-/CD19^-/CD56^-/CD66b^-/EpCAM^+$, 0.04%) and circulating hybrid cells ($CD45^+/CD3^-/CD19^-/CD56^-/CD14^+/EpCAM^+$, 0.2%) were identified, with the latter being ~5X more abundant than the former....” (**Line 380-385**).

Reviewer #3

Major concerns:

32. The proposed model of the development of macrophage-prostate cancer hybrid cells is predominantly based on a highly artificial *in vitro* model. Although three prostate cancer cells are included in the studies, most *in vitro* work is done in the C2-4 prostate cancer cells with the lower CD47 expression and development of hybrids. Moreover, predominantly one model of human macrophages is used, the human U937 derived macrophage like cells. M-CSF stimulated monocytes and murine IC-21 derived macrophages are only used for a limited number of studies. Hybrid cells of primary macrophages with C2-4 cells are rarely found, while none are formed with DU-145 and 22RV1 cells (Fig 1e). It would be very helpful if critical findings could be confirmed in an additional model of prostate cancer and macrophages.

Response: Following the reviewer's comments, we have since increased the scope of our research by conducting additional *in vitro*, *in vivo*, and clinical studies. In this revised manuscript, murine prostate cancer cell lines RM-1 and TRAMP-C2 have been included in our co-culture studies, showing that low Cd47-expressing TRAMP-C2 cells were more susceptible to forming hybrid cells than high Cd47-expressing RM-1 cells (**Supplementary Fig. 5, Line 277-279**). Moreover, in the syngenic mouse models injected with these murine lines, TRAMP-C2-metastasized tumors in the lung were shown to contain more naturally formed hybrid cells (**Fig. 5g-i, Line 281-283**). We have included 94 tissue sections from 89 patients with primary and/or metastatic prostate tumors for a clinical study. The IHC result showed that hybrid cells were preferentially found in metastatic tumors, tumors with high Gleason scores, and patients with older age (**Fig. 5j-p, Line 293-300**).

33. The concept that fusion cells with an abnormal chromosome distribution contribute to cancer development is a long-standing theory, of which supporting data was summarized in a 2008 *Nat Rev Cancer* publication. However, this field of research remains an *in vitro* exercise for decades, apart from a few small cohorts of cancer patients where occurrence of circulating fusion cells was associated with survival (including the present manuscript). In the current manuscript the presence of hybrids in 11 primary prostate cancer was assessed by scRNA sequencing (multiple macrophage and epithelial markers). This is strong, but the question remains what the relevance is of the macrophage-prostate cancer hybrid cells in human prostate cancer? Do they occur in primary prostate tumors and metastases? How big is the interpatient and inpatient variation? Is their presence associated with outcome? Although supporting the *in vitro* findings, the presented translational studies are weak, due to low number of patients and definition of compared cohorts. A relation between multiplex immunohistochemistry staining for markers of hybrid cells in a large cohort of primary and metastatic prostate cancer lesions (TMA), in relation to clinical outcomes (time to metastases, biochemical recurrence, progression free survival) would significantly enhance the relevance of the studies.

Response: This is an important comment regarding the clinical significance of hybrid cells. We have included the 2008 *Nat Rev Cancer* paper in our citation. Two sets of clinical data have since been added to our revised manuscript. First, IHC was performed on 50 primary and 44 metastatic prostate tumors. As described in Response #32, the finding revealed that hybrid cells were preferentially found in metastatic prostate tumors relative to primary tumors, those with higher Gleason scores, and patients of older ages (**Fig. 5j-p**). In addition, *in silico* analysis of RNA-seq datasets from 634 tumor and control samples in four metastatic castration-resistant prostate cancer (mCRPC) cohorts suggests that hybrid cells resemble an mCRPC subtype (**Fig. 6, Supplementary Fig. 7c, d**). Accordingly, we revised the manuscript by describing these new findings in the main text (**Line 293-300 and 316-333**).

Minor concerns:

34. High CD47 expression is associated with advanced cancers and suppressed phagocytosis by myeloid cells. How do the authors explain the association between low CD47 expression and a shorter time to biochemical recurrence in the TCGA dataset (Figure 1a)? Moreover, there seems to be no difference in expression of CD47 between benign prostate tissue and primary prostate cancer, while expression is lower in metastatic prostate cancer. This finding seems to be very consistent in three individual datasets (Figure 1b). How is this explained?

Response: The observation that both primary prostate tumors and benign hyperplastic samples had comparable CD47 expression levels in the three cohorts shown in the revised **Fig. 1a** can partially be validated by *in silico* scRNA-seq analysis of another prostate cancer cohort (n = 15). In the revised legend of **Supplementary Fig. 6f-h**, we have stated that "...**g** Ridgeline plots displayed two peaks, one with and the other without detectable CD47, in epithelial cells of each tissue sample. **h** Violin plots showing CD47 expression of the epithelial cells

(peak 2) in Group 1 and Group 2 cancerous tissues. Higher frequencies of hybrid cell formation tended to occur in primary tumors with lower CD47 levels (Group 1) than tumors with higher levels (Group 2) in the CD47-expressing subpopulations. The four non-cancerous tissues all had comparable expression levels of CD47, and yet the lower incidence of hybrid cell formation than that of Group 1 tumors. One probable explanation is fewer macrophage infiltrations in non-cancerous tissues than in adjacent and tumor sites (see **Fig. 5b-middle-right**).” We have since observed increased infiltration of macrophages in an adjacent site next to a metastatic liver lesion in a mouse model. Therefore, the revised main text has stated that “...this increased macrophage infiltration additionally contributes to hybrid cell formation in low CD47-expressing cancer cells.....” (**Line 273-274**).

35. The claim that macrophages and prostate cancer cells fuse to a hybrid, is based on expression of a single macrophage and an epithelial marker. This claim can be made a lot stronger. Some cytogenetic analysis is presented in Suppl Fig 2d, but how many chromosomes do the macrophage like cells, the prostate cancer cells and the hybrids have?

Response: In the legend of revised **Supplementary Fig. 4d**, we have stated that “...Karyotypes of a parental C4-2 cell, a parental U937 cell, and two CWC cell lines. Complex karyotypes were presented in CWC cells (n = 8 analyzed) with a modal chromosome number ranging from 50 to 62, while C4-2 cells (n = 4) had 86-87 and U937 cells (n = 4) had 55-57 chromosomes...” As shown in **Supplementary Fig. 4e**, hybrid cells acquired marker chromosomes from both parental cells.

36. It should be clear in all figures/legends, which model of macrophages has been used.

Response: We have since revised the panels and relabeled cell line names in **Fig. 1d-g**, **Supplementary Fig. 1d-g**, and the corresponding figure legends.

37. It is surprising that the biggest effect of blocking CD47 (B6H12) on hybrid formation is in the cells with the lowest CD47 expression (C4-2). Suppl Fig 1G. how is this explained?

Response: One possibility is that the dose of B6H12 we applied can further weaken “don't-eat-me” signals in low CD47-expressing cancer cells, while a greater dose may be needed to show the effect of hybrid cell formation in high CD47-expressing cancer cells. For a future study, we will conduct a dose-dependent assessment of different cell lines.

38. Number of genes assessed to establish M1 and M2 polarization of the macrophages is really limited. No single gene expression confirms differentiation unequivocally. Montovani et al described multiple markers of M1 and M2 polarization. The claim that the hybrid cells adopt both polarizations should be substantiated with a wider panel of markers.

Response: We thank the reviewer's comment. Our macrophage polarization genesets were reconstructed based on the recent publications from 2015-2021, including human M1- (n = 25) and M2-like (n = 21) genesets and murine M1- (n = 25) and M2-like (n = 18) genesets. We have since reviewed Dr. Montovani's publications and organized polarization genes from their latest publications from 2010-2020. More than half of our curated genesets were included in their publications. These genesets we used are sufficient to distinguish the distribution patterns between hybrid and parental tumor regions (see **Fig. 2g, h**). When M1- and M2-like indices were recalculated using the combination of their markers (e.g., *IRF3*, *STAT1*, *YM1*, *FIZZ1*, *CDH1*, *JMJD3*, and others), we found only a minor difference between their genesets (see below) and our genesets (see **Fig. 2g, h**).

We sincerely apologize for not including Dr. Montovani's publications as references initially. We have since cited Dr. Montovani's papers in the revised manuscript (See Supplementary Table 1: PMID: 20856220, 31530089, 23766387, 22378047, 26699615, 31178859, 32039007, 34789834, and 31554795). Moreover, we have also validated

eight macrophage markers in animal models and two macrophage markers in human blood samples using CyTOF, respectively (**Fig. 4** and **Fig. 8**)

39. AR signaling is the single most important growth signal in prostate cancer. All three prostate cancer cell lines assessed express AR and are responsive to testosterone. Moreover, also macrophages express AR. Do the hybrid cells express AR? Studies into the functionality of AR signaling in the hybrid cells would be really interesting and relevant.

Response: We appreciate the reviewer's insightful comment regarding the contribution of hybrid cells to prostate cancer biology. Accordingly, we have reanalyzed our spatial transcriptomics data, conducted an *in vitro* assay, and performed *in silico* analysis of four mCRPC cohorts. These new data are presented in **Fig. 6** and **Supplementary Fig. 7**. In the revised Abstract, we have stated that "...Furthermore, hybrid cells intrinsically express low AR/AR-targets, resembling an adenocarcinoma-immune subtype of metastatic castration-resistant prostate cancer (mCRPC)..." (**Line 38-40**). Along with this new finding, we have devoted a new Result section describing how hybrid cells contribute to mCRPC, particularly for double-negative prostate cancer (**Line 302-333**).

40. Also, the rare putative macrophage-prostate cancer cells hybrids in human circulation (0,21% of PBMCs) are identified by one immune cell (and not macrophage) marker (CD45) and one epithelial marker (EpCAM). Considering the limited number and nonspecific markers and its very low frequency, these findings might concern other hybrids of immune and epithelial cells, but also be an artifact.

Response: We apologize for the confusion and have re-plotted the CyTOF data to improve the clarity of **Fig. 8** and **Supplementary Fig. 8**. Immune cells, CTCs, and hybrid cells have been classified based on a CyTOF panel of seven CD markers, three epithelial markers, and two prostate cancer markers (**Fig. 8c**). In the revised text, it has been stated that ".....We then used CyTOF to determine the frequency of hybrid cells in PBMCs of 16 patients. CyTOF profiling of ~2.9 million PBMCs identified ten main populations based on differential expression patterns of different myeloid and epithelial markers (**Fig. 8b, c, Supplementary Fig. 9a, b, Supplementary Table 7**). Two minor populations - CTCs (CD45⁻/CD3⁻/CD19⁻/CD56⁻/CD66b⁻/EpCAM⁺, 0.04%) and circulating hybrid cells (CD45⁺/CD3⁻/CD19⁻/CD56⁻/CD14⁺/EpCAM⁺, 0.2%) were identified, with the latter being ~5X more abundant than the former. CTCs appeared to express higher levels of two additional epithelial markers, CK19 and MUC-1, and two AR targets, PSMA and PSA, than those of circulating hybrid cells (**Supplementary Fig. 9b, c**)..." (**Line 379-387**).

Reviewer #4

41. In lines 120 to 122. The author suggested that “do not eat me signals upon treatment with CD24- Siglec 10, and PDL1-PD1 does not exert an apparent effect compared to CD47-SIRPalpha in modulating the hybrid cell formation”. The question is, what specific molecular changes are induced by CD47-SIRPalpha that make eat me, not process, specifically dominant in high-grade prostate cancer cells?

Response: Among many “don’t-eat-me” signaling pathways, our data suggest that weak CD47-SIRP α signaling plays a role in hybrid cell formation based on three co-culture models (**Fig. 1c-g, Supplementary Fig. 1i-k**). The low activity of forward signaling (i.e., CD47 as a ligand) is insufficient to repel phagocytosis, allowing macrophages to engulf cancer cells. However, the reverse signaling (i.e., SIRP α as a ligand) “*interferes with this process by partially activating a pro-survival function, leading to incomplete digestion of the entrapped cells*” (**Line 131-132**). In the revised manuscript, we have also provided the evidence that “*...Increased BCL-2 expression might enhance anti-apoptotic activities, thus preventing the complete digestion of the entrapped cancer cells inside macrophages*” (**Line 119-121**).

42. In lines 127 to 129, the author suggested that an “increased amount of bcl2 protein may enhance the apoptotic activities to prevent the complete digestion of entrapped cancer cells inside the macrophages”, Did the authors compare the extent of BCL increase in other PCa cell types post-exposure with CD24- Siglec 10, and PDL1-PD1?

Response: While our present study focuses on CD47-SIRP α signaling, we thank the reviewer for this insightful suggestion regarding a potential increase in BCL-2 by other “don’t-eat-me” signals. We plan to conduct a thorough investigation on this matter in future studies.

43. Additionally, Oxidative stress increase or the ross level increase can also influence the severity of the disease. I wonder if the authors evaluated If the CD47-SIRPalpha influences oxidative stress, which could potentially explain the extent of differences between the effects of CD24-Siglec 10, and PDL1-PD1 and CD47-SIRPalpha.

Response: One of our objectives in the manuscript is to describe the contribution of BCL-2-mediated prosurvival function to hybrid cell formation. As suggested by the reviewer, a potential increase in oxidative stress, such as ROS, may also enhance the event. This new direction of investigation on oxidative stress requires full attention in the future.

44. In lines 144 to 147, the authors suggested that “the hybrid cells evasion can influence the immune surveillance and gaining hyperproliferative growth in the host.” For this, the authors used nude mice. Though they admit that the nude mice lack the essential components of the immune system like T and B cells, they used these murine models to focus on macrophage polarization. The concern arises when authors use the outcomes from these models to generalize them to make claims on the influence of their findings on the overall immune surveillance, which is not true. It is strongly recommended to either remove these claims or use C57Bl6 mice grafted with TRAMP2 cells as models to generate hybrid cells.

Response: Following the reviewer’s suggestion, we have used TRAMP-C2 and RM-1 cells for inoculation into immunocompetent C57BL/6J mice. We have reported the new finding in the revised manuscript, stating that “*...Therefore, two syngeneic prostate cancer cell lines, RM-1 and TRAMP-C2, were used for additional tail vein injections into immunocompetent mice. RM-1 cells had high Cd47 expression and showed a lower rate of hybrid cell formation in vitro compared to low Cd47-expressing TRAMP-C2 cells (Supplementary Fig. 5a-h). Then, we compared their rates of hybrid cell formation in vivo. Although invading RM-1 cells formed large lesions in the lung, the incidence of hybrid cell formation was low (~2%), reflecting the in vitro finding (Fig. 5d-f). In contrast, invading TRMAP-C2 cells developed micrometastasis in the lung but displayed hybrid cells in ~30% of the lesions.*” (**Line 276-283**).

45. In line, 158 to 160 author suggested that “to determine the transcriptomic composition of hybrid cells and their impact on host immune microenvironment they perform special expression profiling on hybrid end Parental tumor sections.” Once again, the murine models used in this study are limited because they are insufficient to depict anything in the tumor microenvironment. Secondly, in the transcriptomic analysis, the author suggested that “the observed human and murine genes such as testing the macrophage infiltration.” In these tumor areas, did the authors compare the difference in the infiltration rate or the type of M1/M2 macrophages from the murine models in the context of the cell types (non-hybrid versus hybrid) that were grafted?

Response: In response to the comment, we have re-plotted four spatial expression profiles of murine and human M1- and M2- macrophage genes (**Fig. 2g, h, j, k**). Based on these new data, we found that hybrid tumor regions had lower expression levels of murine M1- and M2-like genes than those of parental tumor regions. Contrarily, the parental tumor mirrored the opposite scenario, showing the enrichment of murine M1- and M2-like genes (**Line 174-179**).

46. In line 195 to 197, the authors suggested that “hybrid cells are likely benefited from the quasi macrophage characteristic to evade further the surveillance of other immune cells in the host”. The question is, did the authors explore any specific aspects **experimentally**, highlighting that the quasi-macrophage characteristics enhance the evasion of MDSCs and natural killer cells?

Response: In addition to showing the decreased infiltration of MDSCs and NK cells in the hybrid tumor relative to the parental tumor (**Supplementary Fig. 3i, j**), we have additionally conducted immunological evasion assays to explore the influence of hybrid cells on other immune cells, particularly T cell subpopulations (**Fig. 3h** and **Supplementary Fig. 4f**). As suggested in the previous manuscript, c-Myc was shown to contribute to quasi-macrophage characteristics to evade immune surveillance. Here we have found that reduced *MYC* via shRNA knock-down in hybrid cells could unmask this immunoevasive camouflage, generating a more anti-tumorigenic microenvironment for these cells (**Line 214-225**).

47. In the segment, “hybrid cells acquire C-MYC modulated m1 to m2 gene transcription to achieve distinct dissemination,” did authors evaluate the efficacy of C-MYC inhibition in reducing the functional capabilities (reducing the *bcl2* expression of hybrid cells), specifically in the context of other, do not eat me modulators such as CD24- Siglec 10, and PDL1-PD1 and CD47-SIRPalpha.

Response: Our current focus is on elucidating c-Myc-mediated immunoevasive features of hybrid cells. We thank the reviewer for providing this new direction to explore the mechanism of anti-phagocytic function. For our future study, we can determine how c-Myc inhibition attenuates BCL-2 activities via CD47-SIRP α signaling. In addition, we will extend the analysis to other “don’t-eat-me” signaling pathways (e.g., CD24-Siglec 10, and PDL1-PD1) and assess how their influences enhance pro-survival BCL-2 in cancer cells to resist phagocytic attack.

48. Also, did the authors compare the changes in the peaks in the c-Myc binding regions in low-grade versus high-grade prostate, cancer cells, or the patients?

Response: Based on a model we proposed in **Fig. 3i**, increased c-Myc binding occurs in M1- and M2-like loci in hybrid cells. This binding may not be observed in parental macrophages. If high-grade tumors (e.g., Gleason Score 10) harbor more hybrid cells than low-grade tumors, we may find enhanced binding peaks of c-Myc in M1- and M2-like loci and other target genes in hybrid cells of the former for a future study. Because tumors usually have heterogeneous cell populations, we could employ single-cell CUT&Tag to differentiate the c-Myc binding patterns of hybrid cells from those of non-hybrid cells in a tumor in our future study.

49. In line 236 to 239 author suggested that “to determine the impact of hybrid cells on metastasis, they isolated the cells from spine and femur fragments of the xenograft”. The questions are (1) Did the authors use the spine and femur from hybrid cells, C42B, 22RV1, and DU145 cells? (2) Did the authors evaluate the changes (if any) in the c-Myc expression in these?

Response: Since 22Rv1 and DU145 co-culture models generated very low numbers of hybrid cells, we have only focused on assessing the metastatic potential of C4-2 hybrid cells in xenograft mice. Indeed, we found ~1% of C4-2 hybrid cells in spine or femur fragments of these mice (**Fig. 4a-c**). An attempt to conduct molecular characterizations (e.g., c-Myc expression changes) of this small cell population was technically challenging. Instead, we performed CyTOF analysis of a macrometastatic lesion. The result demonstrated that increased c-Myc was associated with enhanced epithelial-mesenchymal plasticity and quasi-polarization macrophage state in hybrid cells distantly disseminated to an organ (**Fig. 4i, j**).

50. In the segment “Low CD47 expressing primary tumors are more prone to generate hybrid cells”, did the authors evaluate if the artificially generated hybrid cells could act as a modulator to induce the prevalence of naturally occurring hybrid cells? Secondly. Did the authors compare the extent of hybrid cells that are naturally occurring by grafting them in mice with non-hybrid or hybrid cells from C42B, 22RV1, and DU145 cells?

Response: This first question is very interesting. While it remains to be determined whether laboratory-generated hybrid cells exert a paracrine effect to induce fusion events *in vivo*, host macrophages may partially engulf these cells again in a natural setting, forming “the hybrid of hybrids” like giant macrophage-like cells as shown in **Fig. 8a**. This new area of research deserves full attention for our future research.

We initially subcutaneously inoculated “non-hybrid” parental C4-2 cells into Nu/Nu mice. IHC analysis revealed that 1-2% of hybrid cells were present in xenograft tumors (**Fig. 5a**). This low incidence of hybrid cell formation was probably a result of less filtration of host macrophages into subcutaneous tumor xenografts. In the revised text, we have further stated that “*However, hybrid cells became highly prevalent in a metastatic liver lesion when C4-2 cells invaded the tissue through the bloodstream in tail vein-injected Nu/Nu mice (Fig. 5b-left). Compared to normal tissue, an adjacent uninvolved liver site displayed intense infiltration of singular F4/80⁺ macrophages with a probable F4/80⁺pan-CK⁺ hybrid cell cluster (Fig. 5b-middle-right). Interestingly, hybrid cell clusters became apparent in more than 30% of a metastatic liver site (Fig. 5c). We suggest that this increased macrophage infiltration additionally contributes to hybrid cell formation in low CD47-expressing cancer cells.*” (**Line 266-274**).

“Non-hybrid” parental 22Rv1 or DU145 cells were also used to generate subcutaneous xenografts. However, we could not detect hybrid cells in xenograft tumors by IHC, likely attributed to high CD47 expression levels of these cancer cells that repel phagocytic attacks. Since the finding may not provide any new formation, we have decided not to include these negative data in the revised manuscript.

51. The authors mentioned that a customized Seurat script was used, and 30 principle components were used in the analysis. Is the code readily available for analysis in an RMarkdown file or R script? Also, did the authors optimize the analysis for 30 PC selection and resolution tuning?

Response: The code of our customized Seurat script has been placed in GitHub for public access (https://github.com/zhangz3/TMH_NC/; **Line 886-887**). An initial elbow plot has shown that the top 15-20 principal components (PCs) can capture the majority of signals (see below). To increase the selection stringency suggested (https://satijalab.org/seurat/articles/sctransform_vignette.html), the top 30 PCs have further been chosen for further analysis when using sctransform for data normalization. See **Line 671-673**.

52. Hallmark gene sets were used, and then customized gene sets from the literature. Could the authors have used some of the other immune-specific gene sets within MSigDB?

Response: We used the Hallmark genesets in MSigDB to analyze the gene enrichment of immune and oncogenic pathways. In addition, genesets for M1- and M2-like macrophages were curated through the literature search (see also our responses to Reviewer #3 in Question 38).

53. The authors mentioned using GSE176031 from the GEO repository and filtering out cells of low quality. How many cells were filtered out with their criteria, and how many were kept?

Response: For single-cell RNA-seq data analysis, we have stated in the revised text that “...cells with low quality (i.e., less than 500 unique features or 300 transcripts, or a mitochondrial level over 20%) were filtered out. DoubletFinder was also used to remove doublets. As such, we removed 12,761 cells and kept 22,084 cells for subsequent analysis” (**Line 843-846**).

54. Line 761 authors mentioned using FindAllMarkers in Seurat and the corresponding p-value is given by Bonferroni correction, but this should be the p_val_adj. Also, was there any consideration given to FC when looking at the differentially expressed genes?

Response: Indeed, the corresponding p-value is given by Bonferroni correction. We have also used `logfc.threshold=0.25` and `min.pct=0.25` to confirm the identification of differentially expressed genes.

Reviewer #5

55. The results in this study were mainly generated based on prostate tumor, but the authors didn't mention this point in the abstract and introduction. Do the conclusions inferred from prostate tumor also apply to other types of cancer?

Response: In response to the reviewer's comment, we have added several descriptions regarding the unique role of hybrid cells in prostate cancer in the Abstract and Introduction (Line 38-40 and 66-74). While hybrid cells have previously been "reported in colorectal, pancreatic, lung, and renal cancers and melanoma" (Line 73-74), our present finding further infers that "hybrid cells resemble a subtype of metastatic castration-resistant prostate cancer (mCRPC)" (Line 302-333).

56. In Figure 2b, how did the authors define the minor regions (1a, 3a and 8a)? It would be better to label these minor regions to t-SNE map. The number of spots in these regions are small, especially for 1a, it would be better to discuss the robustness of defining these minor regions.

Response: Based on the reviewer's suggestion, we have replotted the t-SNE map with the minor groups clearly labeled, as shown in new Fig. 2b. Although the number of spots in 1a, 3a, and 8a is small, the robustness of the finding can further be confirmed by differentially expressed genes in these minor regions.

57. Line 174-176 says 'hybrid tumor usually expressed higher human genes (e.g., VIM, TUBB, and MYC) and lower murine genes (e.g., Vim, Col1a1, and Ctsb) compared to the parental tumor', but Figure 2d/e just showed the expression in a subsection but not the whole region, whether is there significant difference between these two tumors on the overall level?

Response: Spatial expression maps of representative regions (parental tumor regions #1, 5, 6; hybrid tumor regions #4 and 7) have been replotted, along with new violin plots showing expression levels of all regions and p-values between parental and hybrid tumors. Significant differences were shown between these two tumors. (see new Fig. 2d, e, and Supplementary Fig. 3a-d).

58. In Figure 3e, the authors should do the statistical test between DMSO and F4 in level of replicate samples, like Figure 3d. Different replicate samples are likely to be considered as the independent variables. Figure 4i/j, Figure 6 and Figure 7 regarding comparison of gene expression between different groups would also be updated.

Response: Additional violin plots of three independent biological replicates are provided on the right. However, we have decided not to present these data for the current manuscript following the comments by Reviewer #2 (i.e., Question 28) and Reviewer #4 (i.e., Question 47). More experiments are needed to extensively prove the influence of c-Myc on hybrid cell formation using F4 or other c-Myc inhibitors for our future study.

CyTOF analysis was performed on 19,257 single cells of one thoracic lesion in our metastatic mouse model (Fig. 4f-m), ~100,000 single cells from PBMCs of two mice (Fig. 7), and 2,878,799 single cells from PBMCs of 16 prostate cancer patients (Fig. 8). Due to the limitation of available tissue or blood samples from our murine models and prostate cancer patients, we have acknowledged that the sample size of these studies is relatively small for generalization of our findings "... Although the sample size of our present murine and human studies is limited, this CyTOF-based liquid biopsy can be useful in determining the prognostic value of hybrid cells in large patient cohort in the future." (Line 459-462).

59. Figure 3H indicated the role of c-Myc in enhancing the transcription of genes related to macrophage polarization by ChIP. It would be better to validate these results by qPCR or Western Blotting as a supplementary file.

Response: We have since conducted two sets of RT-qPCR and one capillary Western immunoassay (WES) to strengthen our description regarding c-Myc-mediated immunoevasive characteristics of hybrid cells (see Fig. 3e-g).

60. In Line 279, the authors stated that putative hybrid cells were identified based on the co-expression of epithelial and macrophage marker genes, so how to choose the threshold to determine a cell which expresses epithelial genes or macrophage genes, the detailed procedure of selecting putative hybrid cells from scRNA-seq data should be added in the Methods. The putative hybrid cells are formed by two cells, did these cells have relative higher gene numbers or umi counts?

Response: Following the reviewer's suggestion, we have provided a detailed procedure in the revised Methods section "...Since no datasets were available to build a classifier model for hybrid cells, those "cancer cells" displaying high levels (top 1%) of macrophage gene signatures and "macrophages" displaying epithelial gene signatures were selected as potential hybrid cells. These hybrid cells had slightly higher UMI (unique molecular identifier) counts or gene numbers than their parental cancer cells or macrophages." (Line 866-870).

61. Following the above point, how do the authors prove that the defined hybrid cells from scRNA-seq is the real hybrid cells but not the doublets, as the doublets is very common in scRNA-seq data? Did the authors try methods of rare cell type identification, do those real hybrid cells show specific gene programs?

Response: scRNA-seq data have been processed to remove the doublets as described in the revised Methods section (Line 844-845). We used "DoubletFinder" to predict doublets based on each real cell's distance in PCA space to artificial doublets created by averaging transcription profiles of randomly chosen cell pairs (McGinnis et al., 2019, *Cell Systems* 8, 329–337). Based on the barplot below, hybrid cells had a very low percentage of predicted doublets, similar to other cell types.

62. In Figure 1D, the authors used CD86 (M1 markers) as the marker of putative hybrid cells. However, in Figure 7A, they used CD163 (M2 markers) to identify hybrid cells. What is the reason for this inconsistency?

Response: To be consistent with Fig. 1d, we have replaced the IF images (originally shown in Fig. 7a) with CD86 as an example (see revised Fig. 8a-middle).

63. There are some redundant plots in figures, it would be good to move them to supplementary figure. For example, figure 1e and 1f.

Response: The redundant plots of flow cytometry results shown in an original figure panel have been moved to Supplementary Fig. 1g.

64. Lines 176-178, the authors stated that "Four of the seven main regions (i.e., #1, 5, 6, and 8) in the parental tumor contained high levels of murine genes while all hybrid tumor regions expressed elevated human genes (Fig. 2f)", it looks that the % of human genes in region 5 of parental tumor is higher, and also the region 1, is it possible to provide the p values for the comparison?

Response: We apologize for the vague description in the original manuscript. As suggested by the reviewer, we have included p-values and differences to compare the human or murine gene expression between each region in the parental tumor to the whole hybrid tumor (see Supplementary Fig. 3e, f).

REVIEWERS' COMMENTS

Reviewer #2 (Remarks to the Author):

As noted in the initial assessment of this manuscript, this is an outstanding paper that provides fresh insight on the fusion theory as a potential explanation for metastasis. The current version has effectively responded to the questions and feedback I provided in the initial evaluation. While there are some minor points I would make, none should prevent the paper from being published in NC. It would be beneficial for the authors to employ the term Tumor Hybrid Cell (THC) to denote this hybrid population that is now more extensively recognized in relation to metastasis.

Minors

1. Line 32: Avoid the use of parenthesis.
2. Line 40: Avoid the use of parenthesis.
3. In figure 1d: Authors must show the percentage of events at least in the last dot blot where the hybrids are shown. In addition, in this dot blot (up-right quadrant) there several subpopulations (with different EGFP levels), authors must discuss the meaning of them (line 95).
4. Although we have no doubts about the quality of the data and given that many of the analyses are based on flow cytometry identification of rare subpopulations, it is necessary to show the gating strategy that has been followed.
5. There is an apparent inconsistency between the data shown in Figure 1e and Supplementary Figure 1d. The former shows a higher frequency of hybrids when C4-2 is used, while the latter shows a more defined and abundant hybrid population when DU145 is used. Perhaps this is due to a visual assessment error, so it is convenient to show the percentages in the panels of Supplementary Figure 1d.
6. The remarkable difference between the frequency of hybrid formation obtained with a monocyte cell line and the primary culture of these cells should be discussed a little more critically. We must remember that in clinical reality the cell that is present is the monocyte/macrophage not a cell line.

7. The authors talk about the generation of five cell lines from the hybrids, however, only data on the characteristics of two of them are shown. On the other hand, they do not clearly show the metastasis hallmarks that should be met such as: the migration capacity of these lines (a simple trans well experiment in comparison with tumor lines and monocytes/macrophages) or the evasion of the lymphocyte response by exposing them to PBMC and measuring the proliferation of CD4 and CD8 in the presence of a mitogen. Some of these can be inferred from the experiments shown in Figure 3h but the assays shown have other aim.

8. Line 210 “These hybrid cell lines exhibited proliferative capacity in long-term culture similar to those previously reported”. However, authors did not show supporting data.

9. We encourage the authors to use the term Tumor Hybrid Cells instead of hybrids and thus begin to establish the acronym THC to identify this important population.

Reviewer #3 (Remarks to the Author):

The questions and concerns raised were well addressed

Reviewer #4 (Remarks to the Author):

The authors have successfully addressed the concerns raised. I have no more comments.

Reviewer #5 (Remarks to the Author):

Most of the comments have been addressed and this revised manuscript has been improved.

Point-by-point response

Reviewer's Comments:

Reviewer #1 (Remarks to the Author)

[none]

Reviewer #2 (Remarks to the Author)

As noted in the initial assessment of this manuscript, this is an outstanding paper that provides fresh insight on the fusion theory as a potential explanation for metastasis. The current version has effectively responded to the questions and feedback I provided in the initial evaluation. While there are some minor points I would make, none should prevent the paper from being published in NC. It would be beneficial for the authors to employ the term Tumor Hybrid Cell (THC) to denote this hybrid population that is now more extensively recognized in relation to metastasis.

Minors

1. Line 32: Avoid the use of parenthesis.

Response: We have revised the text to "*Here we report that the reverse signaling using CD47 as a receptor additionally....*" (Line 29).

2. Line 40: Avoid the use of parenthesis.

Response: We have deleted the parenthesis and revised the text to "*...metastatic castration-resistant prostate cancer. Therefore, phagocytosis-generated THCs may represent a potential target for treating the disease.*" (Line 37-38).

3. In figure 1d: Authors must show the percentage of events at least in the last dot blot where the hybrids are shown. In addition, in this dot blot (up-right quadrant) there several subpopulations (with different EGFP levels), authors must discuss the meaning of them (line 95).

Response: We have added the percentage of events in **Fig. 1d** following the guidelines of *Nature Communications*. We revised the text: "*... However, a CD86⁺/TiV⁺ subpopulation was found to express diverse levels of EGFP⁺, likely attributed to various extents of incomplete phagocytosis...*" (Line 89-90).

4. Although we have no doubts about the quality of the data and given that many of the analyses are based on flow cytometry identification of rare subpopulations, it is necessary to show the gating strategy that has been followed.

Response: We have attached additional gating/sorting strategies in **Supplementary Fig. 10**. All other gating maps are presented in **Fig. 1d, Supplementary Fig. 1d & 1n, Supplementary Fig. 4f, and Supplementary 5b & 5e**.

5. There is an apparent inconsistency between the data shown in Figure 1e and Supplementary Figure 1d. The former shows a higher frequency of hybrids when C4-2 is used, while the latter shows a more defined and abundant hybrid population when DU145 is used. Perhaps this is due to a visual assessment error, so it is convenient to show the percentages in the panels of Supplementary Figure 1d.

Response: We have added the percentage of events in the gating maps in **Fig. 1d** and **Supplementary Fig. 1d**. The percentage of tumor hybrid cells (THCs) is based on the proportion of each population. Most DU145 cells were not engulfed by macrophages, hence resulting in a lower percentage of THCs.

6. The remarkable difference between the frequency of hybrid formation obtained with a monocyte cell line and the primary culture of these cells should be discussed a little more critically. We must remember that in clinical reality the cell that is present is the monocyte/macrophage not a cell line.

Response: We agree with the reviewer's comment and revised the text accordingly "*....An increase in THC formation was observed in cancer cells co-cultured with differentiated macrophages relative to those with monocytes, likely reflecting a higher ingestion capacity of the former than the latter and thus increasing the chance of forming THCs (Fig. 1e, Supplementary Fig. 1e). Moreover, THC formation was increased when macrophages derived from the oncogenic U937 monocyte cell line were used in co-culture relative to those differentiated from primary monocytes (Fig. 1e). We speculated that phagocytosis could be less robust in U937 macrophages than in primary macrophages, resulting in increased incomplete digestion and THC formation. Compared to U937 macrophages, primary macrophages appeared to digest C4-2 cancer cells more efficiently and led to a more rapid decrease in the proportion of these cells, reducing from 40% at 2 hours to 2% at 72 hours in co-culture (Supplementary Fig. 1f)....*"(Line 95-105).

7. The authors talk about the generation of five cell lines from the hybrids, however, only data on the characteristics of two of them are shown. On the other hand, they do not clearly show the metastasis hallmarks that should be met such as: the migration capacity of these lines (a simple trans well experiment in comparison with tumor lines and monocytes/macrophages) or the evasion of the lymphocyte response by exposing them to PBMC and measuring the proliferation of CD4 and CD8 in the presence of a mitogen. Some of these can be inferred from the experiments shown in Figure 3h but the assays shown have other aim.

Response: We thank the reviewer for providing this idea. The focus of this manuscript is to demonstrate how tumor-macrophage hybrid cells acquired c-Myc during the fusion process for immune evasion and distant dissemination. We have demonstrated metastatic features of THCs in animal models and clinical samples (**Fig. 5**). We understand the importance of studying the migration mechanisms of THCs *in vitro*. However, it could be out of the scope of the current study. Following the reviewer's insightful comment, our future study will include a detailed study of metastatic mechanisms in THCs.

8. Line 210 "These hybrid cell lines exhibited proliferative capacity in long-term culture similar to those previously reported". However, authors did not show supporting data.

Response: The proliferative capacity of THCs can be revealed by their ability to form cell spheres in 3-dimensional culture, and these cells have been cultured for over five passages in our laboratory. To clarify the point, we re-arranged the panels of **Supplementary Fig. 4** and revised the text that "*...authenticated their dual macrophage-epithelial characteristics by*

immunofluorescent and karyotypic analyses (Supplementary Fig. 4b-d). These THC lines exhibited proliferative capacity, forming cell spheres in vitro similar to those previously reported (Supplementary Fig. 4e)..." (Line 207-210).

9. We encourage the authors to use the term Tumor Hybrid Cells instead of hybrids and thus begin to establish the acronym THC to identify this important population.

Response: We thank the reviewer's important suggestion and have used the term tumor hybrid cells (THCs) in the title, the main text, the supplemental information, and the point-by-point response file.

Reviewer #3 (Remarks to the Author)

The questions and concerns raised were well addressed.

Reviewer #4 (Remarks to the Author)

The authors have successfully addressed the concerns raised. I have no more comments.

Reviewer #5 (Remarks to the Author)

Most of the comments have been addressed and this revised manuscript has been improved.